# Catalytic properties of trivalent rare-earth oxides with intrinsic surface oxygen vacancy

Kai Xu [1,8], Jin-Cheng Liu[2,3,8], Wei-Wei Wang [1], Lu-Lu Zhou[1], Chao Ma [4], Xuze Guan[5], Feng Ryan Wang [5] ✉, Jun Li [2,6] ✉, Chun-Jiang Jia [1] ✉ & Chun-Hua Yan[7]

Oxygen vacancy ($O_v$) is an anionic defect widely existed in metal oxide lattice, as exemplified by $CeO_2$, $TiO_2$, and ZnO. As $O_v$ can modify the band structure of solid, it improves the physicochemical properties such as the semiconducting performance and catalytic behaviours. We report here a new type of $O_v$ as an intrinsic part of a perfect crystalline surface. Such non-defect $O_v$ stems from the irregular hexagonal sawtooth-shaped structure in the (111) plane of trivalent rare earth oxides ($RE_2O_3$). The materials with such intrinsic $O_v$ structure exhibit excellent performance in ammonia decomposition reaction with surface Ru active sites. Extremely high $H_2$ formation rate has been achieved at ~1 wt% of Ru loading over $Sm_2O_3$, $Y_2O_3$ and $Gd_2O_3$ surface, which is 1.5–20 times higher than reported values in the literature. The discovery of intrinsic $O_v$ suggests great potentials of applying RE oxides in heterogeneous catalysis and surface chemistry.

Oxygen vacancy ($O_v$)[1–3] is a ubiquitous anionic point defect in transition and *f*-element metal oxides. Commonly, the defect type of $O_v$ requires cations with changeable multiple valence states such as $Ce^{3+/4+}$ [4–6] and $Ti^{3+/4+}$ [7,8]. This defect is realized during the treatment in high temperature[9] or reductive conditions[8,10]. The lattice or surface $O^{2-}$ are taken away together with the reduction of metal cations while keeping the crystal structure, leaving $O_v$ within the lattice. On the one hand, the role of $O_v$ in heterogeneous catalysis has been widely reported[4–8]. For example, the generation of $O_v$ in $CeO_2$ (cubic fluorite structure, $Fm\bar{3}m$ space group) accompanied by the reduction of $Ce^{4+}$ to $Ce^{3+}$ changes its surface charge distribution and creates electrophilic sites[11], which plays a crucial role in improving catalytic performance in CO oxidation, water-gas shift and $CO_2$ reduction[5,6,9,10]. On the other hand, such redox process

is hardly possible with irreducible oxides such as $ZrO_2$, $SiO_2$ and $Al_2O_3$, thereby preventing the generation of defect-based $O_v$[12–15].

In this work, we discover a new type of surface $O_v$ that does not require a point defect formation nor the redox of metal cations. Such intrinsic $O_v$ stems from the natural atomic arrangements in certain crystalline surface of rare earth oxide ($RE_2O_3$). We have analysed the structure and surface charge distribution of $RE_2O_3$ (such as $Sm_2O_3$, $Y_2O_3$ and $Gd_2O_3$) with body-centred cubic structure (*Ia3* space group) based on density functional theory (DFT). An irregular hexagonal sawtooth-shaped structure is found in the (111) surface of those $RE_2O_3$, forming intrinsic surface $O_v$. Next to the $O_v$ are penta-coordinated $RE^{3+}$ with strong electrophilic nature. These $RE_2O_3$ with intrinsic surface $O_v$ are loaded with Ru clusters as active metal ($Ru/RE_2O_3$, RE = Y, Sm and

[1]Key Laboratory for Colloid and Interface Chemistry, Key Laboratory of Special Aggregated Materials, School of Chemistry and Chemical Engineering, Shandong University, Jinan 250100, China. [2]Department of Chemistry and Engineering Research Center of Advanced Rare-Earth Materials of Ministry of Education, Tsinghua University, Beijing 100084, China. [3]Center for Rare Earth and Inorganic Functional Materials, School of Materials Science and Engineering & National Institute for Advanced Materials, Nankai University, Tianjin 300350, China. [4]College of Materials Science and Engineering, Hunan University, Changsha 410082, China. [5]Department of Chemical Engineering, University College London, Roberts Building, Torrington Place, London WC1E 7JE, UK. [6]Fundamental Science Center of Rare Earths, Ganjiang Innovation Academy, Chinese Academy of Sciences, Ganzhou 341000, China. [7]Beijing National Laboratory for Molecular Sciences, State Key Lab of Rare Earth Materials Chemistry and Applications, PKU-HKU Joint Lab in Rare Earth Materials and Bioinorganic Chemistry, Peking University, Beijing 100871, China. [8]These authors contributed equally: Kai Xu, Jin-Cheng Liu. ✉e-mail: ryan.wang@ucl.ac.uk; junli@mail.tsinghua.edu.cn; jiacj@sdu.edu.cn

Gd) for ammonia decomposition reaction, in which an optimal N-binding strength is required. These Ru/RE$_2$O$_3$ catalysts exhibit excellent catalytic performance that is comparable to the most active Ru/CeO$_2$ catalyst that is equipped with the defect O$_v$[16], despite that the RE cations are not redox active. During the reaction, the intrinsic O$_v$ has desired absorption strength of NH$_3$ at the neighbouring RE$^{3+}$, and causes Ru species more reducible, which facilitates the initial N–H breaking. Such intrinsic O$_v$ shows promise of utilizing their novel properties of RE$_2$O$_3$ in catalysis, providing suitable adsorption of reaction molecules for oxidation and hydrogenation chemistry.

## Results

### Intrinsic O$_v$ in the RE$_2$O$_3$ surface
Rare earth oxides with cubic structure, such as Sm$_2$O$_3$, Y$_2$O$_3$, and Gd$_2$O$_3$ mainly expose (111) surface at high temperatures or under harsh reaction conditions[5,17]. We found irregular hexagonal sawtooth-shaped structures formed by three 5-coordinated RE atoms and three 4-coordinated O atoms in cubic-phase Y$_2$O$_3$(111), Gd$_2$O$_3$(111) and Sm$_2$O$_3$(111) (Fig. 1 a–c, e–g, i–k). These vacancy structures are slightly different from the surface point defect O$_v$ in CeO$_2$(111), which is surrounded by three 6-coordinated Ce (Supplementary Fig. 1). Three RE-O bonds are broken for each oxygen vacancy, and thus 25% outmost oxygen vacancy are missing on the (111) surface. Similar O$_v$ is also observed in Sm$_2$O$_3$(110) and (100) surfaces (Supplementary Fig. 2). Due to the exposure of unsaturated 5-coordinated RE atoms, these O$_v$ are electrophilic. Such electrophilic sites can adsorb and activate electron-rich molecules, such as NH$_3$ and H$_2$O. The adsorption strength needs to be high enough to form a stable RE-N/O bond and not too high to cause surface poisoning. We have calculated the adsorption energy values of

NH$_3$ molecules on a series of RE$_2$O$_3$ and compared with the standard γ-Al$_2$O$_3$(111) surface (Supplementary Figs. 3–5 and Supplementary Table 1). The adsorption of NH$_3$ on the Sm$_2$O$_3$(111) surface (−0.44 eV) was stronger than the Sm$_2$O$_3$(110) surface (−0.36 eV), but weaker than that on the Sm$_2$O$_3$(100) surface (−0.98 eV). The moderate adsorption of NH$_3$ on Sm$_2$O$_3$(111) surface (−0.44 eV) is more favourable to the activation of NH$_3$ molecule than that on γ-Al$_2$O$_3$(111) surface, because the latter exhibits an excessively strong adsorption of NH$_3$ (−1.74 eV). Besides, in order to illustrate the unique role of intrinsic O$_v$, which is the special spatial structure in catalysts, the adsorption of molecules at different Sm ions on the surface of Sm$_2$O$_3$ is investigated. The adsorption on the vacancy-related Sm sites is stronger than on the non-vacancy Sm sites of Sm$_2$O$_3$(111) surface (Supplementary Fig. 6 and Supplementary Table 1), where adsorption might be too weak for effective catalysis. Thus, intrinsic O$_v$ on the Sm$_2$O$_3$(111) surface showed great superiority in the adsorption of molecules. Ru is a highly efficient active metal to catalyse the ammonia decomposition reaction[18], so we have constructed a supported catalyst model of Ru$_9$ clusters on Sm$_2$O$_3$(111) surface (Fig. 1d, h, l). The NH$_3$ adsorption energy on Ru$_9$/Sm$_2$O$_3$(111) is −0.79 eV, indicating that the addition of Ru promotes the adsorption of NH$_3$ without causing strong binding in Ru$_9$/Al$_2$O$_3$(111) (−1.88 eV). Therefore, RE$_2$O$_3$ support metal catalysts can be promising in catalysing reactions involving electron-rich molecules.

### Structure and catalytic performance of the Ru/RE$_2$O$_3$ catalysts
In order to explore the potential applications of the intrinsic surface O$_v$ in RE$_2$O$_3$, a series of Ru/RE$_2$O$_3$ (Ru/Sm$_2$O$_3$, Ru/Y$_2$O$_3$, Ru/Gd$_2$O$_3$) and reference Ru/Al$_2$O$_3$ catalysts (~1 wt% Ru content, determined by ICP-MS, Supplementary Table 2) were prepared by colloidal deposition

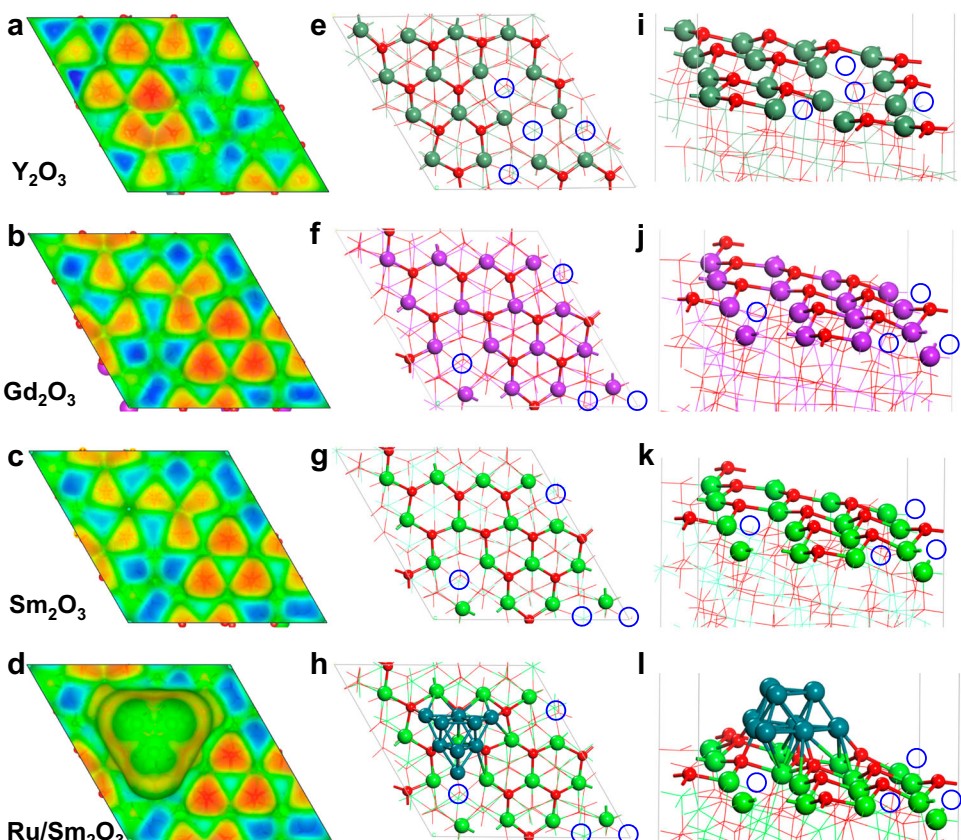

**Fig. 1 | Structure and electrostatic potential results of DFT calculations simulating rare earth oxide (111) surfaces. a–d** Electron density isosurface mapped with electrostatic potential surface of Y$_2$O$_3$(111), Gd$_2$O$_3$(111), Sm$_2$O$_3$(111), and Ru$_9$/Sm$_2$O$_3$(111), respectively, at 0.003 e·bohr$^{-3}$; **e–h** top views, and **i–l** side views of optimized surface structure of Y$_2$O$_3$(111), Gd$_2$O$_3$(111), Sm$_2$O$_3$(111), and Ru$_9$/Sm$_2$O$_3$(111), respectively. Blue circles indicate oxygen vacancy or intrinsic surface O$_v$ structures; blue and orange areas on electrostatic potential surface indicate electrophilic and nucleophilic sites, respectively.

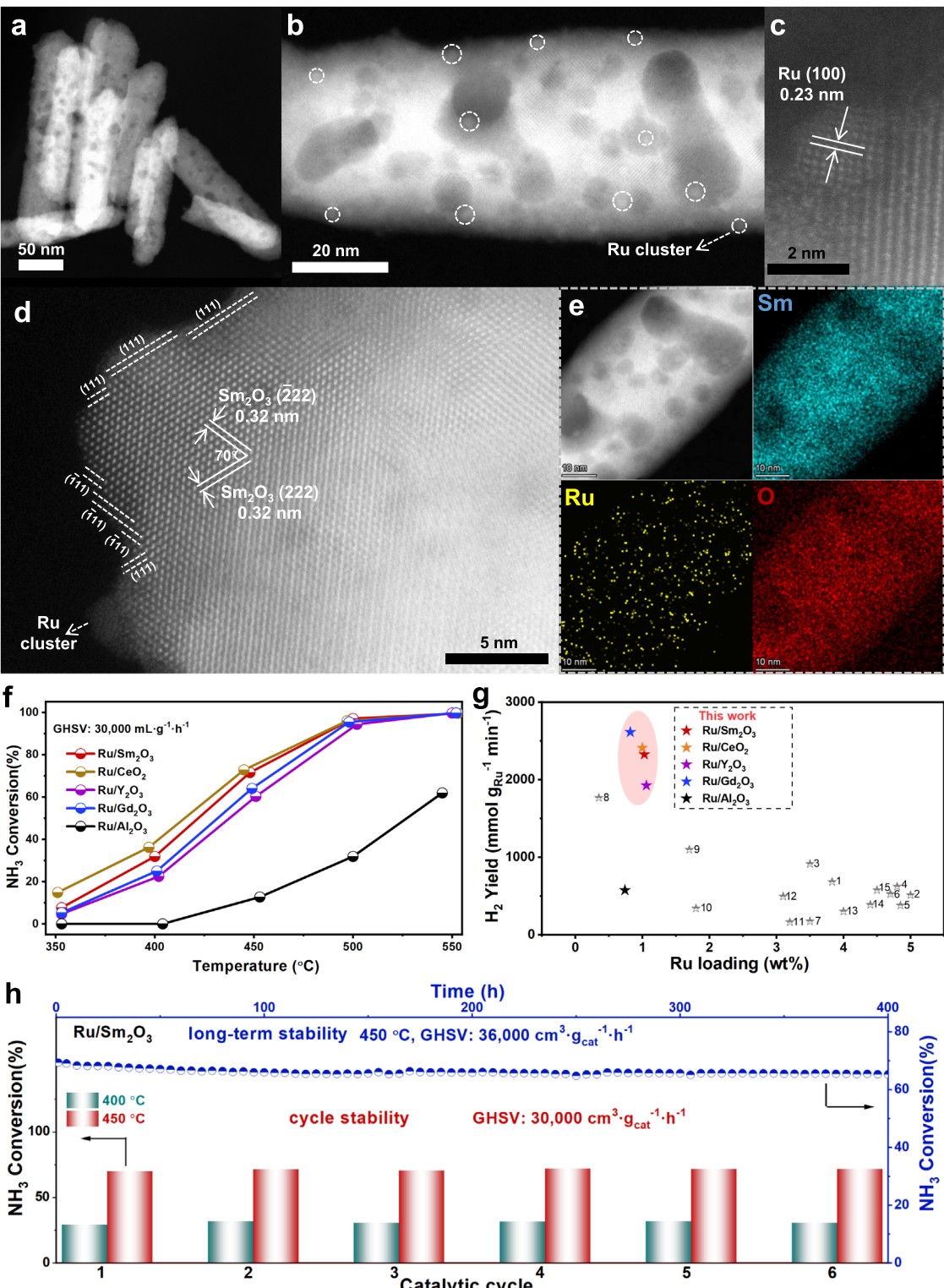

**Fig. 2 | The aberration-corrected HAADF-STEM images and catalytic performance test of catalysts. a–d** The aberration-corrected HAADF-STEM images of the used Ru/Sm$_2$O$_3$; **e** EDS elemental mapping results of the used Ru/Sm$_2$O$_3$; **f** temperature-dependent activities of the catalysts (Ru/Sm$_2$O$_3$, Ru/Y$_2$O$_3$, Ru/Gd$_2$O$_3$ and Ru/Al$_2$O$_3$, GHSV = 30,000 mL·g$^{-1}$·h$^{-1}$); **g** comparison of H$_2$ yield with other Ru-based catalysts: 1: Ru/Sm$_2$O$_3$-p[20], 2:Ru/Y$_2$O$_3$-p[21], 3: K-Ru/MgO[22], 4: K-Ru/CNTs[23], 5: Ru/MgO-CNTs[24], 6: Ru/c-MgO[25], 7: Ru/CaAlO$_x$-w[26], 8: Ru-Ni/CeO$_2$[27], 9: Ru/MgO[28], 10: Ru/LaCeO$_x$[29], 11: Ru/CNF[30], 12: Ru-Cs-Mg/MIL-101[31], 13: Ru/Al$_2$O$_3$[32], 14: Ru-K/ZrO$_2$[18], 15: Ru-K/CNT[18]; (h) the long-term stability (GHSV = 36,000 mL·g$^{-1}$·h$^{-1}$) and the cycle stability test (GHSV = 30,000 mL·g$^{-1}$·h$^{-1}$) of Ru/Sm$_2$O$_3$ catalyst.

method. The transmission electron microscopy (TEM) images of the catalysts were shown in Supplementary Fig. 7. Ru/Sm$_2$O$_3$ and Ru/Gd$_2$O$_3$ had nanorods structure, while Ru/Y$_2$O$_3$ and Ru/Al$_2$O$_3$ had nanosheets and nanoparticles morphology, respectively. The high-angle annular dark-field scanning transmission electron microscopy (HAADF-STEM) images showed that the edge of Sm$_2$O$_3$ (Fig. 2d and Supplementary Fig. 8), Gd$_2$O$_3$ (Supplementary Fig. 9) and Y$_2$O$_3$ (Supplementary Fig. 10) supports after the reaction was mostly {111} plane. The high-resolution

transmission electron microscopy (HRTEM) images (Supplementary Fig. 11–14) and the HAADF-STEM images (Fig. 2a–e, Supplementary Figs. 15 and 16) showed the clear presence of very small Ru clusters on the support. In the HAADF-STEM images of Ru/Sm$_2$O$_3$ (Fig. 2c) and Ru/Al$_2$O$_3$ (Supplementary Fig. 16c, d), we confirmed that the interplanar spacing was consistent with the lattice fringe of Ru (100). According to the particle size distribution (Supplementary Figs. 11c, 12c, 13c, 14c, 17) of all catalysts, the size of Ru cluster in all catalysts was mainly 0.5–2 nm, and the average size was 1.2–1.9 nm. The energy dispersive X-ray spectroscopy (EDS) elemental mappings further confirmed the good dispersion of Ru species on the Sm$_2$O$_3$ (Fig. 2e, Supplementary Fig. 15d) and γ-Al$_2$O$_3$ (Supplementary Fig. 16e, f). The X-ray diffraction (XRD) patterns (Supplementary Fig. 18) of both the fresh and used catalysts only showed the diffraction peak of the corresponding support, which was consistent with the TEM results because the Ru species were highly dispersed with very low loadings.

The catalytic performances of the Ru-based catalysts (Ru/Sm$_2$O$_3$, Ru/Y$_2$O$_3$, Ru/Gd$_2$O$_3$, Ru/CeO$_2$ and Ru/Al$_2$O$_3$) and corresponding pure oxides materials were evaluated in catalytic ammonia decomposition (Fig. 2f and Supplementary Fig. 19a). This reaction was crucial for online H$_2$ production using liquid NH$_3$ as the media for H$_2$ storage and transportation[19]. First of all, pure metal oxide materials had similar and poor NH$_3$ conversion, suggesting a non-catalytic process without the presence of Ru. RE$_2$O$_3$ supported Ru showed similar activities that were obviously higher than the Ru/Al$_2$O$_3$, Ru/TiO$_2$, Ru/SiO$_2$ and Ru/CNTs catalyst (Supplementary Fig. 19b). The NH$_3$ conversion of Ru/RE$_2$O$_3$ catalyst at 450 °C was 60–72%, much higher than that of Ru/Al$_2$O$_3$ catalyst (only ~13%, GHSV = 30,000 mL·g$^{-1}$·h$^{-1}$). The turnover frequency (TOF) value of Ru/Sm$_2$O$_3$ (3.2 s$^{-1}$) was 3 times higher than Ru/Al$_2$O$_3$ (1.1 s$^{-1}$) at 400 °C. The apparent activation energy ($E_a$, Supplementary Fig. 20) of Ru/Sm$_2$O$_3$ (102.2 kJ·mol$^{-1}$), Ru/Y$_2$O$_3$ (123.5 kJ·mol$^{-1}$) and Ru/Gd$_2$O$_3$ (105.1 kJ·mol$^{-1}$) was lower than that of Ru/Al$_2$O$_3$ (136.2 kJ·mol$^{-1}$), exhibiting superiority in catalytic conversion of NH$_3$. Interestingly, even though the $S_{BET}$ of Ru/RE$_2$O$_3$ (Supplementary Table 2) was only 1/3–1/2 of Ru/CeO$_2$[16] ($S_{BET}$ = 106 m$^2$·g$^{-1}$), Ru/RE$_2$O$_3$ catalysts with intrinsic O$_v$ had very similar activity to Ru/CeO$_2$ with defect-based O$_v$. It followed that those intrinsic O$_v$ could have similar effects to the defect-based O$_v$. In addition, the H$_2$ yield with per Ru species of the Ru/RE$_2$O$_3$ catalysts was 1.5–20 times higher than that of other Ru-based catalysts reported[18,20–32] (Supplementary Table 3), proving that the Ru-rare earth oxide catalysts achieved the highest noble-metal atom utilization efficiency for ammonia decomposition reaction at present (Fig. 2g). The durability of Ru/Sm$_2$O$_3$ catalyst was evaluated (Fig. 2h). The NH$_3$ conversion of the Ru/Sm$_2$O$_3$ catalyst decayed by only 4% in 400 h test (GHSV = 36,000 mL·g$^{-1}$·h$^{-1}$). In addition, after six cycles of stability tests, the catalyst still maintained the same NH$_3$ conversion at different temperatures (Supplementary Fig. 21), which also verified the excellent stability. We further tested the performance of the Ru/Sm$_2$O$_3$ catalyst at lower temperatures by the online mass spectrometer (Supplementary Fig. 22). It was found that the catalyst had started catalysing the reaction at a low temperature as 200 °C and had obvious catalytic activity at 250 °C, showing the extraordinary catalytic performance.

The chemical state of the catalyst surface was explored by X-ray photoelectron spectroscopy (XPS, Fig. 3a–c, Supplementary Figs. 23–25 and Supplementary Table 4), Near Edge X-ray absorption fine structure (NEXAFS) (Supplementary Fig. 26) and the in situ infrared (IR) spectroscopy in the transmission mode (Fig. 3d and Supplementary Fig. 27). The initial catalysts contained mainly cationic Ru for all catalysts. After catalysis, most of the cationic Ru in the Ru/Sm$_2$O$_3$ were reduced to metallic Ru, as shown in the XPS result of the used catalyst and the quasi in situ XPS experiment (Fig. 3a). In comparison, after the same treated process, the Ru on Al$_2$O$_3$ remained as oxidative state for the used sample (Fig. 3c), only a small amount of Ru was reduced under quasi in situ XPS measurement condition. Such

comparison suggested that RE$_2$O$_3$ surface with intrinsic O$_v$ helped the reduction of Ru$^{δ+}$ to Ru$^0$ in the presence of H$_2$. In the Sm 3$d$ (Fig. 3b) and Y 3$d$ (Supplementary Fig. 25b) spectra, Sm$^{3+}$ and Y$^{3+}$ were the only species detected, respectively, confirming their non-reducible nature. The surface properties were studied with the Sm M$_{4,5}$ edges and O $K$ edge NEXAFS. Both the fresh and used catalysts had exactly the same Sm$^{3+}$ and O features, suggesting a highly durable surface that maintained the intrinsic O$_v$ (Supplementary Fig. 26). Comparing to the O $K$ edge spectrum of bulk O in Sm$_2$O$_3$ standard[33], the surface O in the Ru/Sm$_2$O$_3$ sample has reduced O 1 s → 5d-π transition comparing to the O 1 s → 5d-σ transition. This showed the slightly different coordination nature between surface and bulk O. To further validate the reduction of Ru, in situ IR experiments during CO adsorption were carried out. The main peak position of Ru/RE$_2$O$_3$ was concentrated between 2040 and 2050 cm$^{-1}$, which was considered as the CO adsorption on Ru$^0$ species[34,35]. While for Ru/Al$_2$O$_3$ catalysts that were concentrated at 2064 cm$^{-1}$, which was considered as the CO adsorption on Ru$^{δ+}$ species (Fig. 3d). The temperature-programmed reduction by H$_2$ (H$_2$-TPR, Supplementary Figs. 28 and 29) results showed that Ru on RE$_2$O$_3$ could be reduced at 123 °C whereas reduction of Ru on Al$_2$O$_3$ required 271 °C. Combining XPS, NEXAFS, in situ IR and H$_2$-TPR, we concluded that Ru over RE$_2$O$_3$ with intrinsic O$_v$ preferred Ru$^0$ whereas Ru over Al$_2$O$_3$ remained at Ru$^{n+}$.

## The catalytic function of RE$_2$O$_3$ with intrinsic O$_v$

The combination of XPS, NEXAFS, IR, DRIFTS and H$_2$-TPR study suggested that Ru over RE$_2$O$_3$ could be easily reduced to Ru$^0$ compared with that of Ru/Al$_2$O$_3$. To validate the catalytic function of RE$_2$O$_3$ with intrinsic O$_v$ in this process and to the NH$_3$ decomposition reaction in general, we first used CO$_2$-temperature programmed desorption (TPD) to investigate the electrophilic nature of the catalyst (Supplementary Fig. 30). Ru/Sm$_2$O$_3$ surface contained more medium base sites compared to Ru/Al$_2$O$_3$, indicating that Ru/Sm$_2$O$_3$ had more effective surface basicity[29]. This medium basicity was conducive to the electron transfer from Sm$_2$O$_3$ to Ru species, and further facilitated the dissociative adsorption of N species at Sm$^{3+}$ site. The results of NH$_3$-TPD (Supplementary Fig. 31) further proved the NH$_3$ desorption was significantly less than that of Ru/Al$_2$O$_3$, which was consistent with the calculation result of adsorption energy values of NH$_3$ molecules on the surface of catalysts (Supplementary Table 1).

To explore the reaction mechanism, we performed first-principles theoretical calculations (Fig. 4) to further study the N−H dissociation. A Ru (0001) slab was adopted to simulate the large-size Ru nanoparticles while Ru$_9$ clusters supported on Sm$_2$O$_3$(111) slab and γ-Al$_2$O$_3$(111) were used to simulate the supported cluster catalyst (Ru/Sm$_2$O$_3$ and Ru/Al$_2$O$_3$). As shown in Fig. 4a, the N-H bond activation barrier on Ru (0001) surface was 1.16 eV, in good agreement with previous work[36]. Comparatively, the activation barrier of N-H bond on Ru$_9$/Al$_2$O$_3$ and Ru$_9$/Sm$_2$O$_3$ was lowered to 0.76 eV and 0.65 eV, respectively. From Fig. 4b, we could see that on both Ru$_9$/Sm$_2$O$_3$ and Ru$_9$/Al$_2$O$_3$, the N-H dissociation went through a synergistic process that NH$_3$ adsorbed on the metal cations in the support surface via the formation N-Sm or N-Al interaction. Meanwhile, H atom in NH$_3$ was captured by the Ru cluster, thus resulting in the N-H bond breaking. So, for Ru$_9$/Sm$_2$O$_3$, the intrinsic O$_v$ sites were responsible for adsorption of NH$_3$, and the interface Ru atoms played the role of activating N-H bonds. After the first N-H bond break, compared to Ru$_9$/Sm$_2$O$_3$ maintained a relatively active state (−1.85 eV), Ru$_9$/Al$_2$O$_3$ was in a very stable state (−3.31 eV), making it difficult for subsequent reactions to occur. As shown in Fig. 4c, the calculated charge density difference showed that electrons transferred from Sm$_2$O$_3$ slab to Ru$_9$ (Supplementary Fig. 32). This result suggested that Sm$_2$O$_3$ was alkalescence and increased the electron density of Ru to dissociate NH$_3$ more favourably. We further calculated the projected density of states (PDOS) of the Ru$_9$/Sm$_2$O$_3$, isolated NH$_3$, and Ru (0001) surface (Fig. 4d). To activate the N-H

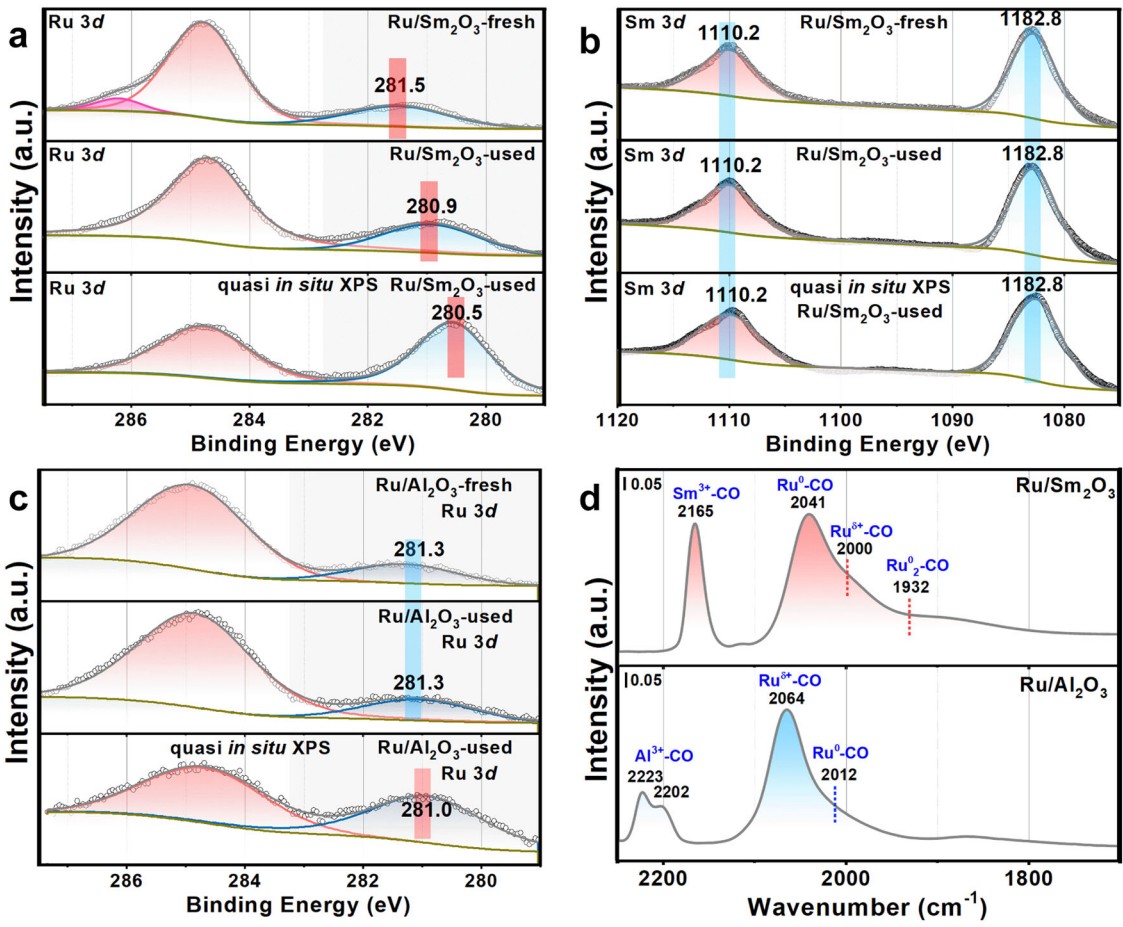

**Fig. 3 | Spectroscopy characterization of catalysts. a–c** XPS results of the fresh and used catalysts: **a** Ru 3$d$ of Ru/Sm$_2$O$_3$, **b** Sm 3$d$ of Ru/Sm$_2$O$_3$, **c** Ru 3$d$ of Ru/Al$_2$O$_3$; **d** the CO adsorption in situ infrared spectroscopy in the transmission mode of used catalysts at 130 K.

bonds, the $d$-band of Ru and LUMO of NH$_3$ must be at similar energy level, which meant Ru metals with higher $d$-band centre will interact with NH$_3$ stronger[37]. The $d$-band centres of Ru$_9$ on Sm$_2$O$_3$ and Ru (0001) slab were −1.37 eV and −1.95 eV, respectively, which accounts for the higher activity of Ru/Sm$_2$O$_3$. The excellent activity of Ru/Sm$_2$O$_3$ was partially from the sintering-resistant property of Sm$_2$O$_3$ substrate. The calculated bind energies of Ru$_9$ with Al$_2$O$_3$ and Sm$_2$O$_3$ were −9.28 eV and −10.24 eV, respectively. Benefiting from the abundant intrinsic surface O$_v$ sites in Sm$_2$O$_3$ surface, Ru clusters were held firmly, thus avoiding the coalescence process and exhibiting solid durability[38,39].

Based on the present results, the effects of the RE$_2$O$_3$ with intrinsic surface O$_v$ structure for catalysis could be summarized as the following. Firstly, the RE$_2$O$_3$ with intrinsic surface O$_v$ promoted the adsorption and activation of the catalyst on Lewis basic reactant molecules to improve the catalytic performance; secondly, the RE$_2$O$_3$ with intrinsic surface O$_v$ enhanced the interaction between the active metals and the rare earth oxide supports, raising the $d$ band of Ru and increase the energy of surface adsorbed NH$_2$. To explore the generality of this effect, we prepared Cu/RE$_2$O$_3$ and Cu/Al$_2$O$_3$ catalysts and tested the activity of the water-gas shift (WGS) reaction (Supplementary Fig. 33), which was also crucial in industrial hydrogen production. The activity of Cu/RE$_2$O$_3$ has been found to be significantly higher than that of Cu/Al$_2$O$_3$, possibly due to the activation of H$_2$O and O-H cleavage via intrinsic O$_v$.

## Discussion

In summary, we have discovered a new type of O$_v$ on the surface of RE$_2$O$_3$. Those O$_v$ stem from the surface symmetric and the atomic

arranges and therefore intrinsic of a crystalline, which is completely different from the conventional defect-based O$_v$. Such RE$_2$O$_3$ with intrinsic O$_v$ is found to play an important role in catalysis, such as ammonia decomposition and WGS reaction. The RE$_2$O$_3$ offers significant advantages, including: (1) moderate adsorption of reaction molecules such as NH$_3$ and H$_2$O; (2) maintaining active species in metallic state, (3) forming unique RE-N(O)-H-Ru configurations for the N(O)-H bond breaking. Such O$_v$-metal synergy is new for those redox inactive metal oxide supports and will bring RE$_2$O$_3$ on the screening system for heterogenous catalysis.

## Methods
### Synthesis of catalysts

The typical synthetic method of Ru colloidal solution has been reported previously[16]. Dissolving 0.15 g RuCl$_3$ (Sinopharm) in 50 mL ethylene glycol (C$_2$H$_6$O$_2$; Sinopharm), and then added 0.16 g NaOH (Sinopharm) to the mixture with constant stirring for 30 min. Next, the solution was refluxed at 160 °C for 3 h. After cooling to room temperature, we obtained the Ru colloidal solution with dark brown.

Sm$_2$O$_3$ nanorod (Sm$_2$O$_3$) and CeO$_2$ nanorod (CeO$_2$) follow the same hydrothermal method. Dissolving NaOH (14.40 g; Sinopharm) in 40 mL deionized water, and then the solution of 3 mmol nitrate (Sm(NO$_3$)$_3$·6H$_2$O (aladdin) and Ce(NO$_3$)$_3$·6H$_2$O (kermel)) was added into the previous mixture and kept stirring for 30 min. Then we transferred the solution to the teflon bottle for hydrothermal reaction at 100 °C for 24 h. The precipitate produced was washed and dried overnight at 60 °C to obtain Sm(OH)$_3$ and CeO$_2$. The Sm(OH)$_3$ was calcined in air at 450 °C for 4 h to obtain Sm$_2$O$_3$.

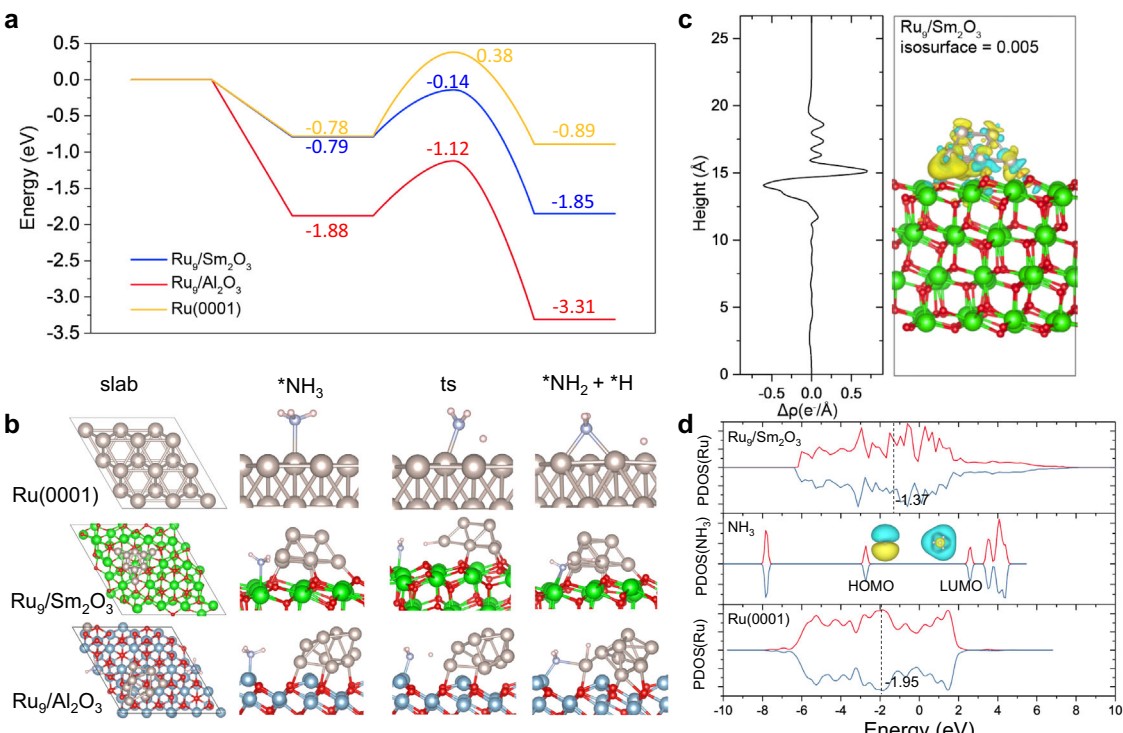

**Fig. 4 | Theoretical results for the reaction mechanism and electronic nature of the active sites. a** Energy profiles and **b** corresponding structures for $NH_3$ adsorption and activation of N–H bond on Ru(0001), $Ru_9/Sm_2O_3(111)$, and $Ru_9/Al_2O_3(111)$, respectively; **c** charge density differences, $\Delta\rho = \rho(Ru_9/Sm_2O_3) - \rho(Ru_9) - \rho(Sm_2O_3)$, for $Ru_9$ adsorption on $Sm_2O_3(111)$ surface (yellow and blue isosurfaces denote where electronic density increase and decrease, respectively) and integration of $\Delta\rho$ in planes parallel to the surface and plotted as a function of the z coordination; (d) the projected density of states (PDOS) of the $Ru_9$ on $Ru_9/Sm_2O_3$, isolated $NH_3$, and Ru(0001) surface (black dash lines label the $d$-band centre of Ru).

$Y_2O_3$ nanosheet was synthesized by hydrothermal method. Dissolving 1.149 g $Y(NO_3)_3 \cdot 6H_2O$ (aladdin) in 60 mL deionized water. And then using 10% NaOH solution adjusted the pH to 12. Then we transferred the solution to the teflon bottle for hydrothermal reaction at 120 °C for 12 h. The precipitate produced was washed and dried overnight at 60 °C. Finally, it was calcined in the air at 500 °C for 6 h.

$Gd_2O_3$ nanorod was synthesized by hydrothermal method. Dissolving 0.02 mol $Gd(NO_3)_3 \cdot 6H_2O$ (aladdin) in 50 mL deionized water. And then using 2.5 mol·L⁻¹ NaOH solution adjusted the pH to 12.8. Then we transferred the solution to the teflon bottle for hydrothermal reaction at 180 °C for 24 h. The precipitate produced was washed and dried overnight at 80 °C. Finally, it was calcined in the air at 450 °C for 2 h.

The Ru-based catalysts were synthesized according to the previous methods[16]. The typical steps of colloidal deposition method were shown as followed. 1 g supports ($Sm_2O_3$, $CeO_2$, $Y_2O_3$, $Gd_2O_3$ and γ-$Al_2O_3$ (commercial, Macklin)) was dissolved and dispersed in 25 mL deionized water, and then added a certain amount of Ru colloid to this mixture and kept stirring for 48 h. Next, ageing for 12 h and the precipitates were collected and washed by centrifugation. The obtained products were dried for 48 h at 60 °C. Finally, the catalysts were reduced in $NH_3$ atmosphere at 550 °C before catalytic test.

## Characterization of catalysts

The inductively coupled plasma mass spectrometer (ICP-MS, PerkinElmer, NexION 350X) analysis was used to detect the actual content of Ru. The $N_2$ adsorption−desorption measurements were on a Builder SSA-4200 analyzer at −196 °C. All the samples were pretreated at 200 °C for 400 min under vacuum. The BET (according to the Brunauer, Emmett and Teeler method) specific surface area ($S_{BET}$) can be calculated from that. The X-ray diffraction (XRD) was carried out on a PANalytical X'pert3 powder diffractometer (40 kV, 40 mA) using Cu Kα

radiation (λ = 0.15406 nm). The diffraction angles (2θ) ranged from 10° to 90°. The thermogravimetric analysis (TGA) was carried out on a simultaneous thermal analyzer (METTLER, TGA/DSC3 + ) in $N_2$. The transmission electron microscopy (TEM) images were conducted on a JEOL JEM-2100F microscope operating at 100 kV. The high-resolution TEM (HRTEM) was carried out under 200 kV on a FEI Talos F200s microscope. The high-angle annular dark-field scanning transmission electron microscopy (HAADF-STEM) images were obtained on a JEOL ARM200F microscope equipped with a probe-forming spherical-aberration corrector and Gatan image filter (Quantum 965). The hydrogen temperature-programmed reduction ($H_2$-TPR) measurements were carried on a Builder PCSA-1000 instrument. After pretreated at 500 °C in air, 50 mg catalysts were reduced by 5%$H_2$/Ar (30 mL·min⁻¹) from room temperature to 700 °C. The $H_2$ consumption was shown by the thermal conductivity detector (TCD). The X-ray photoelectron spectroscopy (XPS) measurements were carried out on an Thermo scientific ESCALAB Xi⁺ XPS spectrometer with Al Kα radiations, and with the C 1$s$ peak at 284.8 eV as an internal standard for all the spectra. The in situ infrared spectroscopy in the transmission mode measurements were conducted in a UHV apparatus combining a FTIR spectrometer (Bruker Vertex 70 v) with a multi-chamber UHV system. The sample was pretreated with $H_2$ in a vacuum at 873 K for 30 min, and then exposed to CO ($10^{-2}$ mbar) at 130 K to collect the spectrogram.

Near Edge X-ray absorption fine structure (NEXAFS) experiments were carried out at the VerSoX beamline (B07-C) of Diamond Light Source (DLS, UK). The beamline has a maximum resolving power hv/Δ(hv) > 5000 with a photon flux > $10^{10}$ photons s⁻¹ from 170 eV to 2000 eV and can be operated (delivering lower flux) up to 2800 eV. The accuracy of the sample and analyser position is typically less than 10 µm. The gas pressure and composition are controlled via a butterfly valve and mass flow controllers. The endstation consists of a fixed

interface flange which holds the entrance cone of the ambient-pressure electron energy analyser (SPECS Phoibos NAP-150). The samples (around 1 mg) were dispersed in water (around 1 mL) and dropped (around 2 droplets) on Au-coated Si (-1 cm × 1 cm), followed by heating at 70 °C to remove the solvent. NEXAFS spectra at Sm $M_4/M_5$ edge and O K-edge were measured in both total electron yield (TEY) mode and Auger electron yield (AEY) mode at room temperature. The measurements were performed under UHV condition.

The temperature programmed desorption (TPD) measurements of the catalysts were performed at the online mass spectrometer (TILON LC-D200M). For the typical $NH_3$-TPD experiments, 300 mg catalysts (20–40 mesh) were pretreated at 550 °C for 1 h in $NH_3$ atmosphere (20 mL·$min^{-1}$). And then it was cooled to room temperature, and held for 1 h in the $NH_3$ atmosphere. Next, switched to Ar and purged until the baseline was stable, collected the signal from room temperature to 800 °C. For the typical $CO_2$-TPD experiments, 300 mg catalysts (20–40 mesh) were pretreated at 550 °C for 1 h in 5%$H_2$/Ar (30 mL·$min^{-1}$). And then it was cooled to room temperature, and held for 1 h in the $CO_2$ atmosphere. Next, switched to Ar and purged until the baseline was stable, collected the signal from room temperature to 800 °C. For the reaction at lower temperatures, 50 mg catalysts (20–40 mesh) were pretreated at 550 °C for 1 h in $NH_3$ atmosphere (20 mL·$min^{-1}$). After cooling down, we collected the signal from 100 °C to 300 °C with a step of 50 °C in continuous pure $NH_3$ flow.

## Theoretical methods and computational details

All static calculations were carried out using spin-polarized density functional theory (DFT) with generalized gradient approximation (GGA) of Perdew–Burke–Ernzerhof (PBE) and PAW pseudopotentials as implemented in VASP 5.4.4 code[40,41]. DFT + U method with U = 4 eV was used to describe the localized rare earth $4f$ states[42]. The valence orbitals were described by plane-wave basis sets with a cutoff energy of 400 eV. Considering the large size of p(4×4)-(111) slabs used in this work, the single gamma-point grid sampling was used for Brillouin Zone integration for geometry optimization, and 3 × 3 × 1 k-mesh was used for density of states calculations. Atomic positions were optimized by using the conjugate gradient algorithm until the forces were less than 0.03 eV/Å. Transition states were searched by climbing image nudged-elastic-band (CI-NEB) method with convergence criterion of 0.05 eV/Å[43,44]. The criterion for electronic self-consistent field convergence was set to $10^{-6}$ eV.

## Catalytic tests

The catalytic performance of the catalysts was tested in a self-constructed fixed-bed flow reactor. The temperature controller (UDIAN, XIAMEN YUDIAN AUTOMATION TECHNOLOGY CO., LTD.) was used in the reactor temperature control system. During the test, 50 mg catalysts (20–40 mesh) mixed with 500 mg quartz sand (20–40 mesh) and then packed into the reactor with an inner diameter of 6 mm. Before the test, the catalysts were activated at 550 °C in pure $NH_3$ atmosphere. Then the activity test was performed from 350 to 550 °C with a step of 50 °C for the reactor temperature (GHSV = 30,000 mL·$g^{-1}$·$h^{-1}$). The outlet gas was analyzed by an online gas chromato-graph (Ouhua GC 9160), and then the real-time $N_2$ and $NH_3$ contents were obtained. The $NH_3$ conversion was calculated through Eq. (1).

$$X_{NH_3} = \frac{2 \times n_{N_2}^{out}}{2 \times n_{N_2}^{out} + n_{NH_3}^{out}} \times 100\% \qquad (1)$$

The stability test of the catalysts was conducted at 450 °C (GHSV = 36,000 mL·$g^{-1}$·$h^{-1}$) for 400 h. The apparent activation energy for the reaction was determined with an equal conversion of 12.5% by tuning the flow rate and temperature.

## Data availability

The main data supporting the findings of this study are available within the article and its Supplementary Information. Additional data are available from the corresponding authors upon request. Source data are provided with this paper.

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

## Acknowledgements

This work was financially supported by National Key Research and Development Program of China (2021YFA1501103), the National Science Fund for Distinguished Young Scholars of China (22225110), the National Science Foundation of China (nos. 22075166, 22271177 and 22033005), the National Science Foundation of Shandong Province of China (ZR2023ZD21 and ZR2023QB187), the Young Scholars Program of Shandong University, EPSRC (EP/P02467X/1 and EP/S018204/2), Royal Society (RG160661, IES\R3\170097, IES\R1\191035, IEC\R3\193038), and by the Guangdong Provincial Key Laboratory of Catalysis (No. 2020B121201002). We thank the Center of Structural Characterizations and Property Measurements at Shandong University for the help on sample characterizations. The calculations were performed by using supercomputers at SUSTech, Tsinghua National Laboratory for Information Science and Technology, and the Computational Chemistry Laboratory of the Department of Chemistry under the Tsinghua Xuetang Talents Program.

## Author contributions

C.-J. J., F. W., J. L. and C.-H. Y. supervised the work; K. X., L.-L. Z. and C.-J. J. designed the experiments, analyzed the results. K. X., J.-C. L., C.-J. J., F. R. W. and J. L. co-wrote the manuscript; K. X. and W.-W. W. performed the in situ DRIFTS and quasi in situ XPS; J.-C. L. and J. L. performed the DFT calculations and analysed the theoretical data; K. X. performed the catalysts preparation, catalytic tests and the TPR tests; C. M. performed the aberration-corrected HAADF-STEM measurements and analysed the results. X. G. and F. R. W. performed the XAFS experiments and analysed the data.

## Competing interests

The authors declare no competing interests.
