## [Peer Review File · Nature Communications]

Catalytic properties of trivalent rare earth oxides with intrinsic surface oxygen vacancyREVIEWER COMMENTS

Reviewer #1 (Remarks to the Author):

In this article, the use of oxides of some rare earth elements with a +3 oxidation state that do not exhibit redox properties is studied and compared with conventional Al_2O_3 and cerium dioxide, where the coexistence of two oxidation states, +3 and +4, facilitates the formation of oxygen vacancies as anionic defects. The idea advocated here is that intrinsic oxygen vacancies in rare earth oxides with a single stable +3 oxidation state, resulting from the reconstruction of {111} facets, are also useful and relevant for catalysis, specifically in the case of ammonia decomposition.

The work includes the characterization of fresh samples and those after the reaction, as well as a DFT study. Despite the potential interest in the study, a fundamental aspect raises questions about its significance and adequacy. This concerns the existence of these intrinsic vacancies due to the reconstruction of {111} facets. The paper shows a HAADF-STEM image of the Ru/ Sm_2O_3 sample with atomic resolution, but similar images are not provided for the other catalysts. Moreover, the Ru nanoparticle in this same image (Figure 1d) has a size larger than reported in the article and is not anchored in the sawtooth reconstruction of {111} planes.

For this study to be convincing, a much more detailed investigation of the crystallographic planes exposed by all catalysts is essential, including a characterization that allows a detailed understanding of the fraction of the reconstructed surface containing intrinsic oxygen vacancies. Without this information, convincing conclusions are impossible. It should also be noted, as extensively discussed in the literature, that nanoforms of rare earth oxides exhibit different crystallographic faces and structural voids, which significantly affect catalytic behavior.

From the images in Figure S5, it can be observed that Sm_2O_3 , in particular, has a large number of voids in the structure of the nanorods. Characterization using TEM tomography or HRTEM or HAADF-STEM with atomic resolution for all samples is necessary, along with a precise assessment of the number of intrinsic oxygen vacancies present. This needs to be carefully measured and presented. Furthermore, catalytic results should also include the Turnover Frequency (TOF) value for a more accurate comparison between different catalysts. Finally, the XPS ratios between Ru and the rare earth oxides should be provided for all fresh and used samples.

Reviewer #2 (Remarks to the Author):

Catalytic properties of trivalent rare earth oxides with intrinsic surface oxygen vacancy by Kai Xu et al.

This is an excellent study combining a range of useful surface science experiments and complementary DFT. Although the DFT approach selected is not state of the art, i.e. not including dispersion, spin orbit coupling, hybrid functionals, or occupancy control of f-orbitals, nevertheless, it looks to be a reliable investigation and we can have confidence in the model. The key finding is that by using a rare earth sesquioxide support, i.e. a RE₂O₃ surface present then the catalytic effects are enhanced. Thus, this is paper and work that is definitely worthy of publication.

My issue is with the focus of the paper, indeed the main comment of the conclusion, that the properties and behaviour are related to the intrinsic oxygen vacancies. Any surface created, will, by definition, have atoms at the surface with a reduced coordination, unless of course, it is further reacted to restore the coordination of the surface atoms. And equally, and for many surfaces some of these atoms that have lost bonds/ligands will be metal atoms. It is well known that metal atoms with a reduced coordination will be more reactive, and as I am interpreting the authors' argument, they are suggesting that the key point is the missing ligand, and the missing oxygen ligand, i.e. the oxygen vacancy, rather than the lower coordinated RE³⁺ ion that is the driving factor. While this is not clear to me, I would not support its publication as is. From the abstract "The discovery of intrinsic Ov suggests great potentials", I am not clear it is a particular discovery, and of itself, I do not see the potential. Clearly, if I have simply misunderstood the concept, perhaps a really clear explanation and some indication on how to exploit this great potential would be answer my concerns.

Reviewer #3 (Remarks to the Author):

The authors have submitted an interesting manuscript on rare earth metal oxides, which are catalytically active for ammonia decomposition. Their main claim is that there are non-defect oxygen vacancies in trivalent RE metal oxides as intrinsic part of the crystal surface and that these sites are responsible for the high activity of Ru supported on these metal oxides for ammonia decomposition. Despite the high number of characterization methods applied and plausible DFT calculations I have a number of concerns with respect to this claim, which have to be addressed by the authors before publication can be recommended.

The surface of a metal oxide is always coordinatively unsaturated, which is the prerequisite for adsorption and catalysis. There is literature on different metal oxide surfaces (e.g., TiO₂ (101)) that

point out the different coordination of surface atoms (five-coordinated and six-coordinated Ti atoms due to a similar sawtooth-shape) AND resulting differences in electronegativity, bond strength, etc. This results in preferred adsorption of gas phase components on these atoms. From my perspective, the observed results can be explained by the differences in adsorption strength and reactivity of the different surface atoms. The question is, why the concept of non-defect oxygen vacancies is introduced and what is the added value of it. I am skeptical if it provides new insights or a new perspective. Moreover, I find the number of provided examples too low for such a generalization. Electron-rich molecules (NH_3 , H_2O) can be adsorbed on electrophilic sites of different origin.

If oxygen vacancies are important for their conclusions, I recommend detecting and quantify the oxygen vacancies. There are papers in literature on the characterization of oxygen vacancies (e.g., K. Le, *TrAC Trends in Analytical Chemistry*, 116 (2019) 102-108), which might help to support the author's claims.

Specific comments:

page 7, bottom: The authors compare their catalysts only with other Ru-based catalysts. How do their catalysts compare with other types of NH_3 decomposition catalysts?

page 8, middle: The authors compare Ru/ RE_2O_3 only with Ru/ Al_2O_3 . How do the results compare for different supports (e.g., TiO_2 , SiO_2 , C, etc.). Maybe their catalysts are only superior compared to alumina, which would cast their conclusions into doubt.

page 8, middle: The authors compare Ru/ RE_2O_3 with much smaller surface area with Ru/ CeO_2 and found similar activities. If two different supports with different surface areas result in similar activities, I would argue that the influence of the support on Ru is negligible or that the nature of support chemistry and surface area balance each other. However, I find it daring to speculate about the influence of oxygen vacancies, whose concentration is unknown and should be determined.

page 8, bottom: The authors claim that the Ru/ RE_2O_3 catalysts show the highest noble-metal atom utilization efficiency for ammonia decomposition, which is not correct, since their own Ru/ Ce_2O_3 catalyst is better according to Table 3.

page 9, bottom: Suppl. Fig. 20c should be 20b.z

Suppl. information:

page S5: Please indicate clearer in Figure caption 2 that CeO₂ is reduced to create a surface vacancy.

page S32, Table 1: How have the adsorption energies been determined? I guess that the adsorption energies are different on the different surfaces. Which surface was used for Ru₉/Sm₂O₃ and Ru₉/Al₂O₃?

Responses to the Reviewers' Comments and the Corresponding Revisions

To Reviewer 1:

Reviewer #1: In this article, the use of oxides of some rare earth elements with a +3 oxidation state that do not exhibit redox properties is studied and compared with conventional Al_2O_3 and cerium dioxide, where the coexistence of two oxidation states, +3 and +4, facilitates the formation of oxygen vacancies as anionic defects. The idea advocated here is that intrinsic oxygen vacancies in rare earth oxides with a single stable +3 oxidation state, resulting from the reconstruction of $\{111\}$ facets, are also useful and relevant for catalysis, specifically in the case of ammonia decomposition.

The work includes the characterization of fresh samples and those after the reaction, as well as a DFT study. Despite the potential interest in the study, a fundamental aspect raises questions about its significance and adequacy. This concerns the existence of these intrinsic vacancies due to the reconstruction of $\{111\}$ facets. The paper shows a HAADF-STEM image of the Ru/ Sm_2O_3 sample with atomic resolution, but similar images are not provided for the other catalysts. Moreover, the Ru nanoparticle in this same image (Figure 1d) has a size larger than reported in the article and is not anchored in the sawtooth reconstruction of $\{111\}$ planes.

For this study to be convincing, a much more detailed investigation of the crystallographic planes exposed by all catalysts is essential, including a characterization that allows a detailed understanding of the fraction of the reconstructed surface containing intrinsic oxygen vacancies. Without this information, convincing conclusions are impossible. It should also be noted, as extensively discussed in the literature, that nanoforms of rare earth oxides exhibit different crystallographic faces and structural voids, which significantly affect catalytic behavior.

From the images in Figure S5, it can be observed that Sm_2O_3 , in particular, has a large number of voids in the structure of the nanorods. Characterization using TEM tomography or HRTEM or HAADF-STEM with atomic resolution for all samples is necessary, along with a precise assessment of the number of intrinsic oxygen vacancies present. This needs to be carefully measured and presented. Furthermore, catalytic results should also include the

Turnover Frequency (TOF) value for a more accurate comparison between different catalysts. Finally, the XPS ratios between Ru and the rare earth oxides should be provided for all fresh and used samples.

Response: Thanks for the reviewer's valuable comments and suggestions. Your feedback has encouraged us to deepen our understanding on these aspects. The above comments mainly include the following aspects.

1. The intrinsic oxygen vacancy structure only exists in {111} planes or not.

“Intrinsic oxygen vacancy” structures are found and analyzed by the theoretical model of RE₂O₃, it is a class of structures that we have discovered rather than a specific structure. Owing to {111} planes of cubic-phase RE₂O₃ being relatively stable at high temperatures or under harsh reaction conditions^{1,2}, it can be regarded as the main model for investigation. Thus, Y₂O₃(111), Gd₂O₃(111) and Sm₂O₃(111) are discussed primarily in this manuscript. We find irregular hexagonal sawtooth-shaped structures formed by three 5-coordinated RE atoms and three 4-coordinated O atoms on the RE₂O₃(111) surface (Fig. 1 a–c, e–g, i–k), which exhibit electrophilic properties.

However, **it does not mean the intrinsic oxygen vacancy only exists in {111} planes, we can also observe similar structures that are electrophilic sites in Sm₂O₃(110) and (100) surface (Supplementary Fig. 2).** As the comments suggested by the reviewer, the different crystal structures in different planes will cause distinction in catalytic reaction activity. In order to comprehensively disclose the effects of different planes on catalytic activity, the adsorption energy of three typical molecules (NH₃, H₂O and O₂) on the various surfaces of Sm₂O₃ are investigated (**Fig. R1-1 and Table R1-1**). The moderate adsorption of NH₃ on the Sm₂O₃(111) surface (–0.44 eV) was stronger than the Sm₂O₃(110) surface (–0.36 eV), but weaker than that on the Sm₂O₃(100) surface (–0.98 eV). These results suggest the Sm₂O₃(111) exhibits moderate adsorption strength among the investigated surfaces, which also demonstrates the rationalization of the theoretical model that we selected. The calculation results of adsorption energy about other molecules, for example, H₂O and O₂ also shows the same trend. In addition, even NH₃ adsorbed on Sm₂O₃(100) surface (–0.98 eV) is still favourable to the activation of NH₃ molecule compared with that on γ -Al₂O₃(111) surface, because the latter exhibits an

excessively strong adsorption of NH_3 (-1.74 eV). The above result suggests the superiority about activation of electron-rich molecules on the surfaces of RE_2O_3 .

The Fig. R1-1 has been added as Supplementary Fig. 4 in the revised supporting information on *page S7*. The Table R1-1 has been added in Supplementary Table 1 in the revised supporting information on *page S38*. The corresponding description has been shown in *page 4, line 89–91* in the revised manuscript and *page S7, line 97–108* in the revised supporting information (highlighted in yellow).

Fig. R1-1 Model of molecules (NH_3 , H_2O and O_2) adsorption on the Sm_2O_3 surfaces. (a) Sm_2O_3 (111) surface; (b) Sm_2O_3 (100) surface; (c) Sm_2O_3 (110) surface.

Table R1-1 The adsorption energy of molecules (NH_3 , H_2O and O_2) on the Sm_2O_3 surface.

Surface	Adsorption energy for NH ₃ (eV)	Adsorption energy for H ₂ O (eV)	Adsorption energy for O ₂ (eV)
Sm ₂ O ₃ (111)	-0.44	-1.52	-1.24
Sm ₂ O ₃ (110)	-0.36	-1.03	-0.71
Sm ₂ O ₃ (100)	-0.98	-2.41	-2.31

2. Further characterization by the HAADF-STEM for the Ru/RE₂O₃ samples.

The intrinsic oxygen vacancy is the precise structure that exists in the lattice planes of RE₂O₃. It is really challenging to obtain the intact and clear images depending on the electron microscope technology. Thus, precisely assessing the number of intrinsic oxygen vacancies by electron microscope is difficult. **In our manuscript, the fact that 25% outmost oxygen vacancy is missing on the (111) surface owing to three RE-O bonds being broken for each oxygen vacancy was clearly described.** This result can reflect the concentration of intrinsic oxygen vacancy on the RE₂O₃ surface.

Next, a much more detailed investigation of the crystallographic planes exposed by Ru/RE₂O₃ (Ru/Sm₂O₃, Ru/Gd₂O₃ and Ru/Y₂O₃) is supplemented. More HAADF-STEM images of Ru/Sm₂O₃ are shown in **Fig. R1-2**. A large amount of Sm₂O₃{111} ((222) planes) can be concluded through the analysis of the lattice fringes. Only a few Sm₂O₃{100} ((400) planes) can be observed in the location of the surface, and no corresponding fringes of Sm₂O₃{110} can be found. Besides, the HAADF-STEM images of the used Ru/Gd₂O₃ and Ru/Y₂O₃ are also supplemented in this response (**Fig. R1-3** and **R1-4**). The images of both the Ru/Gd₂O₃ and Ru/Y₂O₃ mainly expose {111} planes, and the edge of supports (Gd₂O₃ and Y₂O₃) after the reaction is mostly {111} planes. This result further demonstrates that RE₂O₃ mainly exposes {111} surface at high temperatures or under harsh reaction conditions, verifying the model in this manuscript is reasonable.

Fig. R1-2 The aberration-corrected HAADF-STEM images of the used Ru/Sm₂O₃.

Fig. R1-3 The aberration-corrected HAADF-STEM images of the used Ru/Gd₂O₃.

Fig. R1-4 The aberration-corrected HAADF-STEM images of the used Ru/Y₂O₃.

Finally, the size of Ru clusters in Ru/Sm₂O₃ catalyst is mainly 0.5–2.5 nm, and the average size is 1.3 nm (Supplementary Fig. 11). In order to clearly observe the Ru species in **Fig. 2d**, the marked line is deleted (as shown in **Fig. R1-5**). We can clearly observe the morphology of the Ru cluster in such a location on the surface of Sm₂O₃. The Ru species might consist of two types of Ru clusters, which contain the larger clusters with a size of ~1.6 nm and the small clusters with size smaller than 1.0 nm. Besides, it also might be one cluster with irregular morphology, whose longest length is ~3.1 nm. Such a result is still reasonable and matches with the particle size distribution of Ru species. As mentioned above, we cannot directly observe the intrinsic oxygen vacancy in the HAADF-STEM images. Intrinsic oxygen vacancy is the precise structure that exists in the lattice planes of RE₂O₃, whose diameter value is lower than 10 Å. The size of Ru clusters (0.5–2.5 nm) is much larger than that of intrinsic oxygen vacancy. Thus, the Ru clusters are bound to contact with the intrinsic surface oxygen vacancy directly. We mainly focus on the discussion of intrinsic oxygen vacancy-related Ru species in this manuscript. In addition, the voids in the structure of the Sm₂O₃ nanorods are caused by the uneven surface of the support, which also exist in other oxides^{1,3–6}. Such voids cannot affect the main lattice planes and intrinsic oxygen vacancy structure.

Fig. R1-5 The aberration-corrected HAADF-STEM images of the used Ru/Sm₂O₃ in Fig.2.

The Fig. R1-2, R1-3 and R1-4 have been added as Supplementary Fig. 8, 9 and 10 in the revised supporting information on *page S12, S13 and S14*, respectively. The Fig. R1-5 has been added in Fig. 2 in the revised manuscript on *page 7*. The corresponding description has been shown in *page 6, line 122–125* in the revised manuscript and *page S12, line 158–160* in the revised supporting information (highlighted in yellow).

3. The catalytic results should also include the Turnover Frequency (TOF) value for a more accurate comparison between different catalysts.

The catalytic test results of Ru/RE₂O₃ and the RE₂O₃ support suggest that the pure RE₂O₃ showed almost no activity for ammonia decomposition, and Ru species are significant for the high NH₃ conversion. Thus, we calculate the TOF depending on the number of active Ru sites. The dispersity (*D*) of Ru species on the Ru/Sm₂O₃ and Ru/Al₂O₃ are detected by CO chemisorption (**Fig. R1-6**). The test results show that the *D*_{Ru} of Ru/Sm₂O₃ and Ru/Al₂O₃ are 35.3% and 36.9%, respectively.

Fig. R1-6 The CO chemisorption result of Ru/Sm₂O₃ and Ru/Al₂O₃.

Then $n_{\text{effective}}$ of Ru and the TOF is calculated through eq. 1 and 2. The L_{Ru} represents the actual Ru loading determined by ICP-MS, and the M_{Ru} represents the relative atomic mass of Ru (101.07 g/mol). And the TOF values are calculated based on H₂ formation yield at a low NH₃ conversion below 15 %. The calculated result is shown in **Table R1-2**. The TOF of Ru/Sm₂O₃ is 3 times higher than Ru/Al₂O₃ at 400 °C. **The corresponding description has been shown in page 8, line 155–157 in the revised manuscript.** We are looking forward to your next comments.

$$n_{\text{effective}} = \frac{1 \text{ g} \times L_{\text{Ru}} \times D_{\text{Ru}}}{M_{\text{Ru}}} \quad 1$$

$$\text{TOF} = \frac{H_2 \text{ formation yield}}{n_{\text{effective}}} \quad 2$$

Table R1-2 The TOF of the catalyst.

Samples	Ru loading (%)	Temperature (°C)	Dispersity of Ru	H ₂ formation yield (μmolH ₂ ·g ⁻¹ ·s ⁻¹)	TOF (s ⁻¹)
Ru/Sm ₂ O ₃	1.0	360	35.5%	33.5	1.0
		400		113.9	3.2
Ru/Al ₂ O ₃	0.7	400	26.9%	23.2	1.1

4. **The XPS ratios between Ru and the rare earth oxides should be provided for all fresh and used samples.**

The atomic% results of Ru and RE (Sm and Y) are shown in **Table R1-3**, and the semiquantitative results are concluded from the XPS survey. However, the result will be affected by the strong overlap in C 1s and Ru 3d binding energy in XPS, thus, the result of ICP-MS is more accurate. **The Table R1-3 have been added as Supplementary Table 4 in the revised supporting information on page S41.** We are looking forward to your next comments.

Table R1-3 The atomic% results of Ru and RE (Sm and Y) in XPS.

Samples	Atomic%	
	Ru	RE (Sm and Y)
Ru/Sm ₂ O ₃ -fresh	1.0	13.9
Ru/Sm ₂ O ₃ -used	0.9	21.1
Ru/Y ₂ O ₃ -fresh	0.4	26.6
Ru/Y ₂ O ₃ -used	0.7	34.3

To Reviewer 2:

Reviewer #2: *Catalytic properties of trivalent rare earth oxides with intrinsic surface oxygen vacancy by Kai Xu et al.*

This is an excellent study combining a range of useful surface science experiments and complementary DFT. Although the DFT approach selected is not state of the art, i.e. not including dispersion, spin orbit coupling, hybrid functionals, or occupancy control of f-orbitals, nevertheless, it looks to be a reliable investigation and we can have confidence in the model. The key finding is that by using a rare earth sesquioxide support, i.e. a RE₂O₃ surface present then the catalytic effects are enhanced. Thus, this is paper and work that is definitely worthy of publication.

Response: Thanks for the reviewer's valuable comments.

My issue is with the focus of the paper, indeed the main comment of the conclusion, that the properties and behavior are related to the intrinsic oxygen vacancies. Any surface created, will, by definition, have atoms at the surface with a reduced coordination, unless of course, it is further reacted to restore the coordination of the surface atoms. And equally, and for many surfaces some of these atoms that have lost bonds/ligands will be metal atoms. It is well known that metal atoms with a reduced coordination will be more reactive, and as I am interpreting the authors' argument, they are suggesting that the key point is the missing ligand, and the missing oxygen ligand, i.e. the oxygen vacancy, rather than the lower coordinated Re³⁺ ion that is the driving factor. While this is not clear to me, I would not support its publication as is. From the abstract "The discovery of intrinsic Ov suggests great potentials", I am not clear it is a particular discovery, and of itself, I do not see the potential. Clearly, if I have simply misunderstood the concept, perhaps a really clear explanation and some indication on how to exploit this great potential would be answer my concerns.

Response: We greatly appreciate the reviewer's insightful comments and the opportunity to clarify the novelty and significance of our findings regarding the intrinsic oxygen vacancies in

Sm₂O₃. The reviewer's feedback has encouraged us to deepen our understanding on these aspects, which is central to our study's contribution to the field of catalysis.

1. Clarification on the Nature of Intrinsic Oxygen Vacancies in Sm₂O₃.

Intrinsic oxygen vacancies in Sm₂O₃ represent a fundamentally distinct class of vacancies when compared to traditional vacancies observed in oxides like CeO₂. Unlike in CeO₂, where oxygen vacancies result in a change in the oxidation state of cerium from +4 to +3, **the intrinsic oxygen vacancies in Sm₂O₃ are an inherent feature of its crystal structure. These vacancies are present without altering the oxidation state of Samarium, which remains constant at +3.** This unique characteristic leads to a uniform distribution of oxygen vacancies across the crystal surface, providing a different reactivity profile compared to CeO₂.

2. The Unique Role of Intrinsic Oxygen Vacancies in Catalysis.

The influence of particular spatial structures on molecular adsorption should include geometric effect as well as electronic effect. The activation of molecules is generally achieved by electron transfer between the adsorbed molecules on the surface and the active sites, which further causes structural buckling and a longer bond length. The nucleophilic N atoms in the NH₃ are likely to be adsorbed and activated efficiently at the electrophilic intrinsic oxygen vacancy. In addition, the steric hindrance is presented during the adsorption of the NH₃ (triangular pyramidal molecule). Intrinsic oxygen vacancy might provide favorable space for their adsorption and activation, meeting the requirements of their local coordination environment.

The coordination number of Sm ions adjacent to intrinsic oxygen vacancies on the Sm₂O₃(111) surface is reduced to 5 or 6, as opposed to a full coordination number of 7 in bulk Sm₂O₃. This structural aspect has significant implications for catalysis. Owing to the distinction of Sm ions in spatial position, the Sm ions adjacent to intrinsic O_v and those not in intrinsic O_v might exhibit different coordination environments and charge densities. Such differences will cause the distinction in the adsorption for reactant molecules. Our investigation reveals that these intrinsic oxygen vacancies moderate the adsorption energies of essential reactants (NH₃, H₂O, O₂) differently from both traditional oxygen vacancies in CeO₂(111) and intact Sm₂O₃. **The adsorption is not as strong as on CeO₂(111) with vacancies which are too robust and**

make difficulties in product desorption. Conversely, the adsorption of these molecules on the Sm ions adjacent to intrinsic oxygen vacancies is stronger than on non-vacancy Sm sites of $\text{Sm}_2\text{O}_3(111)$ surface, where adsorption might be too weak for effective catalysis (Fig. R2-1, R2-2 and Table 2-1).

This balanced adsorption strength afforded by the intrinsic oxygen vacancies in Sm_2O_3 is what we refer to as having the potentials for catalysis. It facilitates a catalytic process where product desorption is not hindered by overly strong adsorption, thus potentially enhancing catalytic turnover. This moderation in adsorption strength is pivotal for reactions such as ammonia decomposition, which require a delicate balance between adsorption and desorption to achieve optimal efficiency.

The Fig. R2-1 and R2-2 have been added as Supplementary Fig. 6 in the revised supporting information on page S9. The Table R2-1 has been added in Supplementary Table 1 in the revised supporting information on page S38. The corresponding description has been shown in page 4, line 94–100 in the revised manuscript and page S9 and S10, line 120–145 in the revised supporting information (highlighted in yellow). We are looking forward to your following comments.

Fig. R2-1 Model of molecules (NH₃, H₂O and O₂) adsorption on the Sm₂O₃ surface. (a) Sm ion adjacent to intrinsic O_v; (b) Sm ion that not in intrinsic O_v.

Fig. R2-2 Model of molecules (NH₃, H₂O and O₂) adsorption on the CeO₂ with surface vacancy (CeO₂-v (111)).

Table R2-1 The adsorption energy of molecules (NH₃, H₂O and O₂) on the different sites.

Surface	Adsorption energy for NH ₃ (eV)	Adsorption energy for H ₂ O (eV)	Adsorption energy for O ₂ (eV)
Sm ₂ O ₃ (111)	-0.44	-1.52	-1.24
Sm ₂ O ₃ (111)-non- vacancy site	-0.27	-0.28	-0.01
CeO ₂ (111)-O _v	-0.58	-2.71	-2.51

To Reviewer 3:

Reviewer #3: The authors have submitted an interesting manuscript on rare earth metal oxides, which are catalytically active for ammonia decomposition. Their main claim is that there are non-defect oxygen vacancies in trivalent RE metal oxides as intrinsic part of the crystal surface and that these sites are responsible for the high activity of Ru supported on these metal oxides for ammonia decomposition. Despite the high number of characterization methods applied and plausible DFT calculations I have a number of concerns with respect to this claim, which have to be addressed by the authors before publication can be recommended.

Response: Thanks for the reviewer's valuable comments.

The surface of a metal oxide is always coordinatively unsaturated, which is the prerequisite for adsorption and catalysis. There is literature on different metal oxide surfaces (e.g., TiO₂ (101)) that point out the different coordination of surface atoms (five-coordinated and six-coordinated Ti atoms due to a similar sawtooth-shape) AND resulting differences in electronegativity, bond strength, etc. This results in preferred adsorption of gas phase components on these atoms. From my perspective, the observed results can be explained by the differences in adsorption strength and reactivity of the different surface atoms. The question is, why the concept of non-defect oxygen vacancies is introduced and what is the added value of it. I am skeptical if it provides new insights or a new perspective. Moreover, I find the number of provided examples too low for such a generalization. Electron-rich molecules (NH₃, H₂O) can be adsorbed on electrophilic sites of different origin.

If oxygen vacancies are important for their conclusions, I recommend detecting and quantify the oxygen vacancies. There are papers in literature on the characterization of oxygen vacancies (e.g., K. Le, TrAC Trends in Analytical Chemistry, 116 (2019) 102-108), which might help to support the author's claims.

Response: Thanks for the reviewer's valuable comments. The reviewer's feedback has encouraged us to deepen our understanding on these aspects, which is central to our study's contribution to the field of catalysis.

First, the concept of non-defect oxygen vacancies (intrinsic oxygen vacancies) in Sm₂O₃ represents a fundamentally distinct class of vacancies when compared to traditional vacancies observed in oxides like CeO₂. Unlike in CeO₂, where oxygen vacancies result in a change in the oxidation state of cerium from +4 to +3, the intrinsic oxygen vacancies in Sm₂O₃ are an inherent feature of its crystal structure. These vacancies are present without altering the oxidation state of Sm ions, which remains constant at +3. This unique characteristic leads to a uniform distribution of oxygen vacancies across the crystal surface, providing a different reactivity profile compared to CeO₂.

Besides, as the reviewer suggested, the different adsorption of gas might be explained by the different coordination of surface atoms. However, in our opinion, **the influence of particular spatial structures on molecular adsorption should include geometric effect as well as electronic effect.** The activation of molecules is generally achieved by electron transfer between the adsorbed molecules on the surface and the active sites, which further causes structural buckling and a longer bond length. The nucleophilic N atoms in the NH₃ were likely to be adsorbed and activated efficiently at the electrophilic intrinsic oxygen vacancy. In addition, the steric hindrance is presented during the adsorption of the NH₃ (triangular pyramidal molecule) process. Intrinsic oxygen vacancy might provide favorable space for their adsorption and activation, meeting the requirements of their local coordination environment.

Thus, we investigate the adsorption of molecules at different Sm ions on the surface of Sm₂O₃. Besides, in order to provide more examples for such a generalization, the adsorptions of NH₃, H₂O, O₂ are explored. The coordination number of Sm ions adjacent to intrinsic oxygen vacancies on the Sm₂O₃(111) surface is reduced to 5 or 6, as opposed to a full coordination number of 7 in bulk Sm₂O₃. This structural aspect has significant implications for catalysis. Owing to the distinction of Sm ions in spatial position, the Sm ions adjacent to intrinsic O_v and those not in intrinsic O_v might exhibit different coordination environments and charge densities. Such differences will cause the distinction in the adsorption for reactant molecules. Our investigation reveals that these intrinsic oxygen vacancies moderate the adsorption energies of essential reactants (NH₃, H₂O, O₂) differently from both traditional oxygen vacancies in CeO₂(111) and intact Sm₂O₃. **The adsorption is not as strong as on CeO₂(111) with**

vacancies which are too robust and make difficulties in product desorption. Conversely, the adsorption of these molecules on the Sm ions adjacent to intrinsic oxygen vacancies is stronger than on non-vacancy Sm sites of $\text{Sm}_2\text{O}_3(111)$ surface, where adsorption might be too weak for effective catalysis (Fig. R3-1, R3-2 and Table 3-1). This balanced adsorption strength afforded by the intrinsic oxygen vacancies in Sm_2O_3 is what we refer to as having the potentials for catalysis. It facilitates a catalytic process where product desorption is not hindered by overly strong adsorption, thus potentially enhancing catalytic turnover. **This moderation in adsorption strength is pivotal for reactions such as ammonia decomposition, which require a delicate balance between adsorption and desorption to achieve optimal efficiency.**

Fig. R3-1 Model of molecules (NH_3 , H_2O and O_2) adsorption on the Sm_2O_3 surface. (a) Sm ion adjacent to intrinsic O_v ; (b) Sm ion that not in intrinsic O_v .

Fig. R3-2 Model of molecules (NH_3 , H_2O and O_2) adsorption on the CeO_2 with surface vacancy ($\text{CeO}_2\text{-v}$ (111)).

Table R3-1 The adsorption energy of molecules (NH₃, H₂O and O₂) on the different sites.

Surface	Adsorption energy for NH ₃ (eV)	Adsorption energy for H ₂ O (eV)	Adsorption energy for O ₂ (eV)
Sm ₂ O ₃ (111)	-0.44	-1.52	-1.24
Sm ₂ O ₃ (111)-non- vacancy site	-0.27	-0.28	-0.01
CeO ₂ (111)-O _v	-0.58	-2.71	-2.51

Finally, we discuss the quantification of the intrinsic oxygen vacancies. The intrinsic oxygen vacancies in Sm₂O₃ are an inherent feature of its crystal structure. These vacancies are present without altering the oxidation state of Sm ions, which remains constant at +3. Thus, precisely assessing the number of intrinsic oxygen vacancies present by traditional technology depending on the change of valence was difficult. **In our manuscript, the fact that 25% utmost oxygen vacancy is missing on the (111) surface owing to three RE-O bonds being broken for each oxygen vacancy was clearly described.** This result could reflect the concentration of intrinsic oxygen vacancy on the RE₂O₃ surface.

The Fig. R3-1 and R3-2 have been added as Supplementary Fig. 6 in the revised supporting information on *page S9*. The Table R3-1 has been added in Supplementary Table 1 in the revised supporting information on *page S38*. The corresponding description has been shown in *page 4, line 94–100* in the revised manuscript and *page S9 and S10, line 120–145* in the revised supporting information (highlighted in yellow).

Specific comments:

- *page 7, bottom: The authors compare their catalysts only with other Ru-based catalysts. How do their catalysts compare with other types of NH₃ decomposition catalysts?*

Response: Thanks for the reviewer's valuable suggestion. The active metals for ammonia decomposition mainly include Ru, Co, Ni, Fe and so on. A number of reports⁷⁻⁹ have shown

that Ru exhibits the highest ammonia decomposition activity, and the activity order of all metals is Ru > Rh/Co > Ni > Pt > Pd > Fe. Thus, Ru-based catalysts represent the optimal catalyst for ammonia decomposition. According to the reviewer's suggestion, in order to clearly illustrate the superiority of the catalyst, the typical catalysts of other metals are supplemented in **Table 3-2**. **Table R3-2 has been added in Supplementary Table 3 in the revised supporting information on page S40.** We are looking forward to your following comments.

Table 3-2 Comparison of catalytic performances for ammonia decomposition reaction over various catalysts.

Catalysts	Metal loading (wt.%)	GHSV (NH ₃) mL·g _{cat} ⁻¹ ·h ⁻¹)	Yield (mmolH ₂ ·g _{cat} ⁻¹ ·min ⁻¹)	Yield (mmolH ₂ ·G _{metal} ⁻¹ ·min ⁻¹)	Reference
Co/Y ₂ O ₃ (450°C)	10.00	6,000	1.9	19	10
Ni/Y ₂ O ₃ (500°C)	10.00	6,000	3.3	33	11
Co _{0.7} Sm _{0.3} O _x (500°C)	-	-	97.2	-	12
Co/La-MgO(5) (550°C)	20.00	124,000	91.0	455	
Ni/La-MgO(5) (550°C)	20.00	124,000	86.0	430	13
Fe/La-MgO(5) (550°C)	20.00	124,000	60.0	300	
Fe-CNFs/mica (600°C)	-	6,500	7	-	14

➤ *page 8, middle: The authors compare Ru/RE₂O₃ only with Ru/Al₂O₃. How do the results compare for different supports (e.g., TiO₂, SiO₂, C, etc.). Maybe their catalysts are only superior compared to alumina, which would cast their conclusions into doubt.*

Response: Thanks for the reviewer's valuable comments. The catalysts that supported on TiO₂, SiO₂, CNTs are prepared, which show low NH₃ conversion during the test process (**Fig. R3-3**). Besides, a number of reports have investigated the catalytic performance of different supports for ammonia decomposition^{8,15,16}. The catalyst prepared with rare earth oxide, MgO

and C as the support exhibits excellent catalytic activity. The Fig. R3-3 has been added as Supplementary Fig. 19b in the revised supporting information on page S23. The corresponding description has been shown in page 8, line 152–154 in the revised manuscript and page S23, line 229–230 in the revised supporting information (highlighted in yellow). We are looking forward to your next comments.

Fig. R3-3 Temperature-dependent activities of the catalysts with different supports.

Ru/Sm₂O₃, Ru/TiO₂, Ru/SiO₂ and Ru/CNTs, GHSV = 30,000 mL·g⁻¹·h⁻¹.

- page 8, middle: The authors compare Ru/RE₂O₃ with much smaller surface area with Ru/CeO₂ and found similar activities. If two different supports with different surface areas result in similar activities, I would argue that the influence of the support on Ru is negligible or that the nature of support chemistry and surface area balance each other. However, I find it daring to speculate about the influence of oxygen vacancies, whose concentration is unknown and should be determined.

Response: Thanks for the reviewer’s valuable comments. The S_{BET} of catalysts not only affects the adsorption of NH₃, but also plays an important role in the loading of Ru species. The catalytic test results of Ru/RE₂O₃ and the RE₂O₃ support suggest that the pure RE₂O₃ shows

almost no activity for ammonia decomposition, and Ru species are significant for the high NH_3 conversion. Combining experimental results and DFT calculation, we believe that the RE-O_v-Ru interfacial sites are crucial for the ammonia decomposition. In this work, we load low-content Ru owing to its high price and obtained extremely high activity. Although the lower S_{BET} of Ru/RE₂O₃, but it can achieve the Ru loading with low content and highly dispersed, which provides sufficient assurance for catalytic activity. Besides, as mentioned above, the intrinsic oxygen vacancies in Sm₂O₃ are an inherent feature of its crystal structure. These vacancies are present without altering the oxidation state of Sm ions, which remains constant at +3. Thus, precisely assessing the number of intrinsic oxygen vacancies present by traditional technology depending on the change of valence is difficult. In our manuscript, the fact that 25% outmost oxygen vacancy is missing on the (111) surface owing to three RE-O bonds are broken for each oxygen vacancy is clearly described. This result can reflect the concentration of intrinsic oxygen vacancy on the RE₂O₃ surface.

➤ *page 8, bottom: The authors claim that the Ru/RE₂O₃ catalysts show the highest noble-metal atom utilization efficiency for ammonia decomposition, which is not correct, since their own Ru/Ce₂O₃ catalyst is better according to Table 3.*

Response: Thanks for the reviewer's valuable comments. As a traditional rare earth oxide, CeO₂ exhibits almost the same activity as Ru/Sm₂O₃. The difference can be considered within average error due to the test detail. In order to avoid controversy, the expression in this manuscript changes to the Ru-rare earth oxide catalyst from the Ru/RE₂O₃ in the corresponding position. **The corresponding description has been shown in page 8, line 163–166 in the revised manuscript (highlighted in yellow).** We are looking forward to your next comments.

➤ *page 9, bottom: Suppl. Fig. 20c should be 20b.z*

Response: Thanks for the reviewer's valuable suggestion. The wrong have been modified.

Suppl. information:

- *page S5: Please indicate clearer in Figure caption 2 that CeO₂ is reduced to create a surface vacancy.*

Response: Thanks for the reviewer's valuable suggestion. The corresponding sections have been modified in *page S4, line 60–66 in the revised supporting information (highlighted in yellow)* according to the requirements.

- *page S32, Table 1: How have the adsorption energies been determined? I guess that the adsorption energies are different on the different surfaces. Which surface was used for Ru₉/Sm₂O₃ and Ru₉/Al₂O₃?*

Response: Thanks for the reviewer's valuable comments. **Intrinsic oxygen vacancy structures are found and analyzed by the theoretical model of RE₂O₃, it is a class of structures that we have discovered rather than a specific structure.** Owing to {111} planes of cubic-phase RE₂O₃ being relatively stable at high temperatures or under harsh reaction conditions^{1,2}, it could be regarded as the main model for investigation. Besides, the HAADF-STEM images of used Ru/RE₂O₃ (Ru/Sm₂O₃, Ru/Gd₂O₃ and Ru/Y₂O₃, **Fig. R3-4, R3-5 and R3-6**) show they mainly expose {111} planes, and the edge of Ru/RE₂O₃ supports after the reaction is mostly {111} planes. Thus, Y₂O₃(111), Gd₂O₃(111) and Sm₂O₃(111) are discussed primarily in this manuscript. And in order to obtain a clear comparison, Al₂O₃(111) is selected to be explored in this manuscript.

Fig. R3-4 The aberration-corrected HAADF-STEM images of the used Ru/Sm₂O₃.

Fig. R3-5 The aberration-corrected HAADF-STEM images of the used Ru/Gd₂O₃.

Fig. R3-6 The aberration-corrected HAADF-STEM images of the used Ru/Y₂O₃.

In addition, the adsorption energy of three typical molecules (NH₃, H₂O and O₂) on the various surfaces of Sm₂O₃ are investigated (**Fig. R3-7 and Table R3-3**) to comprehensively disclose the effects of different planes on catalytic activity. The moderate adsorption of NH₃ on the Sm₂O₃(111) surface (−0.44 eV) is stronger than the Sm₂O₃(110) surface (−0.36 eV), but weaker than that on the Sm₂O₃(100) surface (−0.98 eV). These results suggest that Sm₂O₃(111) exhibits moderate adsorption strength among the investigated surfaces, which also demonstrates the rationalization of the theoretical model that we selected. The calculation results of adsorption energy about other molecules, for example, H₂O and O₂ also shows the same trend. In addition, even NH₃ adsorbed on Sm₂O₃(100) surface (−0.98 eV) is still favourable to the activation of NH₃ molecule compared with that on γ -Al₂O₃(111) surface, because the latter exhibits an excessively strong adsorption of NH₃ (−1.74 eV). **The above result suggests the superiority about activation of electron-rich molecules on the surfaces of RE₂O₃.**

The Fig. R3-4, R3-5 and R3-6 have been added as Supplementary Fig. 8, 9 and 10 in the revised supporting information on *page S12, S13 and S14*, respectively. The corresponding description has been shown in *page 6, line 122–125* in the revised manuscript and *page S12, line 158–160* in the revised supporting information (highlighted

in yellow). The Fig. R3-7 has been added as Supplementary Fig. 4 in the revised supporting information on *page S7*. The Table R3-3 has been added in Supplementary Table 1 in the revised supporting information on *page S38*. The corresponding description has been shown in *page 4, line 89–91* in the revised manuscript and *page S7, line 97–108* in the revised supporting information (highlighted in yellow).

Fig. R3-7 Model of molecules (NH_3 , H_2O and O_2) adsorption on the Sm_2O_3 surface. (a) Sm_2O_3 (111) surface; (b) Sm_2O_3 (100) surface; (c) Sm_2O_3 (110) surface.

Table R3-3 The adsorption energy of molecules (NH₃, H₂O and O₂) on the Sm₂O₃ surface.

Surface	Adsorption energy for NH ₃ (eV)	Adsorption energy for H ₂ O (eV)	Adsorption energy for O ₂ (eV)
Sm ₂ O ₃ (111)	-0.44	-1.52	-1.24
Sm ₂ O ₃ (110)	-0.36	-1.03	-0.71
Sm ₂ O ₃ (100)	-0.98	-2.41	-2.31

References

1. Zhang, Y. et al. Unraveling the physical chemistry and materials science of CeO₂-based nanostructures. *Chem.* **7**, 2022–2059 (2021).
2. Nolan, M. et al. Density functional theory studies of the structure and electronic structure of pure and defective low index surfaces of ceria. *Surf. Sci.* **576**, 217–229 (2005).
3. Yan, H. et al. Construction of stabilized bulk-nano interfaces for highly promoted inverse CeO₂/Cu catalyst. *Nat. Commun.* **10**, 3470 (2019).
4. Wei, S. et al. The effect of reactants adsorption and products desorption for Au/TiO₂ in catalyzing CO oxidation. *J. Catal.* **376**, 134–145 (2019).
5. Guo, Y. et al. Uniform 2 nm gold nanoparticles supported on iron oxides as active catalysts for CO oxidation reaction: structure–activity relationship. *Nanoscale* **7**, 4920–4928 (2015).
6. Zheng, L. et al. Metal-organic framework derived Cu/ZnO catalysts for continuous hydrogenolysis of glycerol. *Appl. Catal. B: Environ.* **203**, 146–153 (2017).
7. Ganley, J. C.; Thomas, F. S.; Seebauer, E. G.; Masel, R. I. A priori catalytic activity correlations: the difficult case of hydrogen production from ammonia. *Catal. Lett.* **96**, 117–122 (2004).
8. Yin, S. et al. Investigation on the catalysis of CO_x-free hydrogen generation from ammonia. *J. Catal.* **224**, 384–396 (2004).
9. Hansgen, D. A., Vlachos, D. G. & Chen, J. G. Using first principles to predict bimetallic catalysts for the ammonia decomposition reaction. *Nat. Chem.* **2**, 484–489 (2010).
10. Huang, C. et al. Ce_{0.6}Zr_{0.3}Y_{0.1}O₂ solid solutions-supported Ni–Co bimetal nanocatalysts for NH₃ decomposition. *Appl. Surf. Sci.* **478**, 708 (2019).
11. Okura, K. et al. Ammonia Decomposition over Nickel Catalysts Supported on Rare-Earth Oxides for the On-Site Generation of Hydrogen. *ChemCatChem*, **8**, 2988 (2016).
12. Wu, C.-P. et al. Co_aSm_bO_x Catalyst with Excellent Catalytic Performance for NH₃ Decomposition. *Chin. J. Chem.* **39**, 2359 (2021).
13. Hu, X.-C. Transition metal nanoparticles supported La-promoted MgO as catalysts for hydrogen production via catalytic decomposition of ammonia. *J. Energy Chem.* **38**, 41 (2019).

14. Duan, X. *et al.* Tuning the size and shape of Fe nanoparticles on carbon nanofibers for catalytic ammonia decomposition. *Appl. Catal. B Environ.* **101**, 189–196 (2011).
15. Yin, S.-F., Xu, B.-Q., Ng, C.-F. & Au, C.-T. Nano Ru/CNTs: a highly active and stable catalyst for the generation of CO_x-free hydrogen in ammonia decomposition. *Appl. Catal. B Environ.* **48**, 237–241 (2004).
16. Lucentini, I. *et al.* Review of the decomposition of ammonia to generate hydrogen. *Ind. Eng. Chem. Res.* **60**, 18560–18611 (2021).

REVIEWERS' COMMENTS

Reviewer #1 (Remarks to the Author):

The authors have responded in detail to my questions and suggestions and have significantly improved the quality of the article. I only have one final comment, which is that it would be highly advisable, as I previously suggested, for the authors to estimate the percentage of each face present in the samples based on the TEM results. If the authors include this information, my recommendation is for the work to be published.

Reviewer #2 (Remarks to the Author):

I thank the authors for the comments, and response to my initial review. I confess that I struggled with their concept of the intrinsic oxygen surface vacancy, as a surface is not intrinsic, i.e. not in thermal equilibrium, and they are not talking about intrinsic point defects formed thermally at surfaces. As I understand it, these are unoccupied oxygen sites present once the surface is formed. Moreover, such empty surface sites on the surface of Re_2O_3 mimic the behaviour of the CeO_2 surface oxygen vacancy.

This is an interesting finding, and the work looks sounds and of interest to the community. Hence, I recommend acceptance.

Reviewer #3 (Remarks to the Author):

I would like to thank the authors for addressing all inputs of the reviewers with great care. The study is for sure well made and the results are clearly presented. However, in my opinion the main problem with the manuscript still persists. I have noticed that reviewer #2 shares my concern about the concept of intrinsic oxygen vacancies that are used to generalize the specific results. The explanations provided by the authors do not convince me to introduce this concept for the explanation of catalytic activities on perfect phase surfaces without redox-active metals.

It is known that surface atoms are always coordinatively unsaturated, which is needed for adsorption and activation of reactants on the catalyst surface. It is also well known that the phase with the most active surface is reactant- and reaction-specific, since the strength of adsorption must be just right for the desired reaction. This seems to apply for certain sites on the (111) surface of Sm_2O_3 , as convincingly explained in the manuscript. However, the authors compare these sites with redox-active sites on CeO_2 , for which oxygen vacancies are accepted. Although I agree that

the sites claimed by the authors seem to be most suitable ones with just the right adsorption strength, I would not introduce the term "oxygen vacancy" in this context. It is simply the right level of undercoordination of these surface Sm atoms that trigger the reaction.

Similar effects have been described before for on Ti (101) surfaces, which are also not redox-active and expose five- and six-coordinated Ti atoms in a zig-zag surface structure at the atomic scale. Only one type of coordinated Ti proved to be highly active for a specific reaction. (I am sure that one may find many more examples.) The argumentation in this example did not need to introduce "intrinsic oxygen vacancies". The same term could have been used there, but I find it misleading since it should be reserved for redox-active materials, in which the metal is reduced upon removal of oxygen from a surface site. Thus, I think we should keep the established terminology and argue with different levels of coordination to describe the reactivity but not with oxygen vacancies on non-redox metals.

Responses to the Reviewers' Comments and the Corresponding Revisions

Reviewer #1:

Comment: The authors have responded in detail to my questions and suggestions and have significantly improved the quality of the article. I only have one final comment, which is that it would be highly advisable, as I previously suggested, for the authors to estimate the percentage of each face present in the samples based on the TEM results. If the authors include this information, my recommendation is for the work to be published.

Response: Thanks for reviewer's comments. After careful consideration of the reviewers' comments, we supplemented the manuscript, which greatly improved the quality of our research work. The suggestion that estimating the percentage of each face present in the samples based on the TEM results is difficult to achieve for these catalysts. Generally, HRTEM combined with theoretical simulation can be used to estimate the percentage of exposed crystal faces of single crystal samples with the same size, shape and length. However, this process is also very complicated. (Yang, H. G. et al. Anatase TiO₂ single crystals with a large percentage of reactive facets. *Nature* **453**, 638 (2008). Qu J. et al. Determination of Crystallographic Orientation and Exposed Facets of Titanium Oxide Nanocrystals. *Adv. Mater.* **34**, 2203320 (2022).) The quantification of exposed crystal faces in the complex polycrystalline catalysts is difficult because the nano-morphology of the oxide particles is not completely consistent. For example, Sm₂O₃ exhibits the nanorod morphology in TEM images, but the length, size, and edge shape of each nanorod are inconsistent. Thus, the quantification of exposed crystal faces might show a large margin of error. However, we could confirm that rare earth oxides with cubic structures, such as Sm₂O₃, Y₂O₃, and Gd₂O₃ mainly expose (111) surface at high temperatures or under harsh reaction conditions based on the HAADF-STEM results and related literatures. **The corresponding description has been shown in page 4, line 72–74 in the revised manuscript.** Thanks for the reviewer again.

Reviewer #2:

Comment: I thank the authors for the comments, and response to my initial review. I confess that I struggled with their concept of the intrinsic oxygen surface vacancy, as a surface is not intrinsic, i.e. not in thermal equilibrium, and they are not talking about intrinsic point defects formed thermally at surfaces. As I understand it, these are unoccupied oxygen sites present once the surface is formed. Moreover, such empty surface sites on the surface of RE₂O₃ mimic the behaviour of the CeO₂ surface oxygen vacancy.

This is an interesting finding, and the work looks sounds and of interest to the community. Hence, I recommend acceptance.

Response: Thanks for reviewer's comments. The professional and detailed comments of the reviewers were very helpful for us, and the quality of our manuscript has been greatly improved compared to the original version. Thanks for the reviewer again.

Reviewer #3:

Comment: I would like to thank the authors for addressing all inputs of the reviewers with great care. The study is for sure well made and the results are clearly presented. However, in my opinion the main problem with the manuscript still persists. I have noticed that reviewer #2 shares my concern about the concept of intrinsic oxygen vacancies that are used to generalize the specific results. The explanations provided by the authors do not convince me to introduce this concept for the explanation of catalytic activities on perfect phase surfaces without redox-active metals.

It is known that surface atoms are always coordinatively unsaturated, which is needed for adsorption and activation of reactants on the catalyst surface. It is also well known that the phase with the most active surface is reactant- and reaction-specific, since the strength of adsorption must be just right for the desired reaction. This seems to apply for certain sites on the (111) surface of Sm_2O_3 , as convincingly explained in the manuscript. However, the authors compare these sites with redox-active sites on CeO_2 , for which oxygen vacancies are accepted. Although I agree that the sites claimed by the authors seem to be most suitable ones with just the right adsorption strength, I would not introduce the term "oxygen vacancy" in this context. It is simply the right level of undercoordination of these surface Sm atoms that trigger the reaction.

Similar effects have been described before for on Ti (101) surfaces, which are also not redox-active and expose five- and six-coordinated Ti atoms in a zig-zag surface structure at the atomic scale. Only one type of coordinated Ti proved to be highly active for a specific reaction. (I am sure that one may find many more examples.) The argumentation in this example did not need to introduce "intrinsic oxygen vacancies". The same term could have been used there, but I find it misleading since it should be reserved for redox-active materials, in which the metal is reduced upon removal of oxygen from a surface site. Thus, I think we should keep the established terminology and argue with different levels of coordination to describe the reactivity but not with oxygen vacancies on non-redox metals.

Response: Thanks for reviewer's comments. Your professional and detailed comments of the reviewers were very helpful for us. And for your final question, our response is as follows:

1. The influence of particular spatial structures on molecular adsorption should include geometric effect as well as electronic effect. Thus, the superiority of intrinsic oxygen vacancies in molecule activation is not only owing to the different levels of coordination, but also because the special structure is impotent. This structural aspect has significant implications for catalysis. Owing to the distinction of Sm ions in spatial position, the Sm ions adjacent to intrinsic O_v and those not in intrinsic O_v might exhibit different coordination environments and charge densities. Such differences will cause the distinction in the adsorption of reactant molecules. Intrinsic oxygen vacancy might provide favorable space for their adsorption and activation, meeting the requirements of their local coordination environment.

Table R1 The adsorption energy of molecules (NH_3 , H_2O and O_2) on the different Sm sites.

Surface	Adsorption energy for NH_3 (eV)	Adsorption energy for H_2O (eV)	Adsorption energy for O_2 (eV)
$Sm_2O_3(111)$	-0.44	-1.52	-1.24
$Sm_2O_3(111)$ -non-vacancy site	-0.27	-0.28	-0.01

2. The “intrinsic oxygen vacancies” are the active sites on the surface of catalyst for activation of molecules, which is proved by the experimental and theoretical studies in this manuscript. The special structure is shown as irregular hexagonal sawtooth-shaped structures formed by three 5-coordinated RE atoms and three 4-coordinated O atoms. Three RE-O bonds are broken for each vacancy, suggesting the formation of this

structure is related to the deficiency of oxygen. Besides, the local structure with similar geometry and charge distribution to defect oxygen vacancy is owing to the special atomic arrangements in certain specific crystalline surfaces. Based on above analysis, we named such active sites as “intrinsic oxygen vacancies”.

Thanks for the reviewer again. Your careful and professional comments greatly improve the quality of our research work.